# Nurse-led home-visitation programme for first-time mothers in reducing maltreatment and improving child health and development (BB:2-6): longer-term outcomes from a randomised cohort using data linkage

Michael Robling [1,2] Fiona V Lugg-Widger [1] Rebecca Cannings-John,[1] Lianna Angel,[1] Sue Channon,[1] Deborah Fitzsimmons,[3] Kerenza Hood,[1] Joyce Kenkre,[4] Gwenllian Moody,[1] Eleri Owen-Jones [1] Rhys D Pockett [3] Julia Sanders,[5] Jeremy Segrott [1,2] Thomas Slater [6]

For numbered affiliations see end of article.

**Correspondence to**
Professor Michael Robling;
roblingmr@cardiff.ac.uk

## ABSTRACT

**Objectives** Measure effectiveness of family nurse partnership (FNP) home-visiting programme in reducing maltreatment and improving maternal health and child health, developmental and educational outcomes; explore effect moderators, mediators; describe costs.

**Design** Follow-up of BB:0–2 trial cohort (ISRCTN:23019866) up to age 7 years in England using record linkage.

**Participants** 1618 mothers aged 19 years or younger and their firstborn child(ren) recruited to BB:0–2 trial at less than 25 weeks gestation and not mandatorily withdrawn from trial or opted out. Intervention families were offered up to a maximum of 64 home visits by specially trained nurses from pregnancy until firstborn child was 2 years old, plus usually provided health and social care support. Comparator was usual care alone.

**Outcome measures** Primary outcome: state-verified child-in-need status recorded at any time during follow-up. Secondary outcomes: referral to social services, child protection registration (plan), child-in-need categorisation, looked-after status, recorded injuries and ingestions any time during follow-up, early childcare and educational attendance, school readiness and attainment at key stage 1 (KS1), healthcare costs.

**Results** Match rates for 1547 eligible children (1517 singletons, 15 sets of twins) were 98.3% (NHS Digital) and 97.4% (National Pupil Database). There was no difference between study arms in the proportion of children being registered as in need (adjusted OR 0.98, 95% CI 0.74 to 1.31), or for any other measure of maltreatment. Children in the FNP arm were more likely to achieve a good level of development at reception age (school readiness) (adjusted OR 1.24, 95% CI 1.01 to 1.52). After adjusting for birth month, children in FNP arm were more likely to reach the expected standard in reading at KS1 (adjusted OR 1.26, 95% CI 1.02 to 1.57). We found no trial arm differences for resource use and costs.

### Strengths and limitations of this study

- ► We used administrative records and novel data linkage to provide longer-term follow-up for a well-characterised trial cohort.
- ► This has maximised cohort retention and minimised response bias.
- ► Prospectively collected trial data on resource use (eg, participant-reported contact with health visitors) and family nurse partnership service records (eg, nurse home visits) provide an indication of support received although the former lacks some granularity. We map out how this informs the comparison of novel intervention with progressive universal usual care.
- ► We used data from a range of sectors including health, social care and education to assess the broad impact of the specialist home visiting programme.
- ► The nature of data recorded in service records place restrictions on its utility for some analyses.

**Conclusions** FNP did not improve maltreatment or maternal outcomes. There was evidence of small advantages in school readiness and attainment at KS1.
**Trial registration number** ISRCTN23019866.

## INTRODUCTION

It is a UK policy priority to protect children from maltreatment and to promote their healthy development. The family nurse partnership (FNP) is a preventative home-visiting programme for young women expecting their first child. FNP was developed in the USA as the nurse family partnership (NFP) and subsequently adapted for implementation in England in 2007.

Three US trials have demonstrated programme improvements in prenatal health behaviours and birth outcomes, sensitive childcare, maternal life-course and child functioning.[1–3] A subgroup analysis of poor unmarried teenage mothers in the first US trial (n=54) found verified maltreatment by age 2 years in 19% of 32 control children and 4% of the 22 children in the group in receipt of NFP during both pregnancy and infancy (mean % difference: 0.15, 95% CI –0.01 to 0.31).[1] Following up the same trial cohort, adjusted rates of verified state reports of child abuse and neglect perpetrated by mothers were lower in the group visited by nurses in pregnancy and infancy (0.29) when compared with the women in the control group (0.54) by 15 years (p<0.001).[4] Additionally, there was a 56% relative reduction in emergency department encounters for injuries and ingestions during the second year of life. In the Memphis trial of NFP, the number days in hospital following an injury or ingestion in the first 2 years of life (a possible indicator of maltreatment) was lower for children of mothers visited by nurses in pregnancy and infancy than children of mothers in the control group.[2] For children with state-verified maltreatment reported by age 4, children in the NFP group had fewer risks for harm than the control group between 25 and 50 months of life. A trial of VoorZorg, the Dutch adaptation of NFP, found a reduction in child-protecting agency reports by age 3 years for families in the home-visiting arm.,[5]

We evaluated short-term FNP programme outcomes to age 2 years in our Building Blocks trial (BB:0–2).[6] We found no difference for four primary outcomes: maternal tobacco use in late pregnancy, birth weight of the baby, proportion of women with a second pregnancy within 24 months post partum and emergency attendances and hospital admissions for the child within 24 months post partum. There were some differences in favour of FNP for secondary child development outcomes including maternal-reported cognitive function and language at 24 months.[6] As the previous US trials showed benefit for maltreatment outcomes increasingly after age four we sought to establish whether FNP moderates maltreatment outcomes over a medium-term period. Through access to administrative data, we aimed to determine impact across a range of child maltreatment outcomes and key indicators of neglect (eg, injuries and ingestions). We also sought to assess impacts on other programme relevant developmental and maternal outcomes as well as costs.

## Objectives
Primary: to determine the effectiveness of the FNP programme when added to usually provided health and social care in reducing maltreatment, when compared with usually provided health and social care alone. Secondary: to determine programme effectiveness in reducing maltreatment, medium-term programme outcomes such as subsequent pregnancies, school readiness and educational outcomes, the impact of moderators of programme effect and the cost and consequences of the programme.

## METHODS
### Study design, setting and participants
This study followed a cohort of mothers and children who had participated in the BB:0–2 trial for a further 5 years using administrative data only.[7 8] Children and mothers were followed up until the child reached key stage 1 teacher-based assessment (KS1) by which time most children were 7 years old.

Building Blocks: 2–6 (BB:2–6) study participants were women and their first child (or twins, if relevant) who were not mandatorily withdrawn from the BB:0–2 trial; did not electively withdraw and/or did not opt-out of this follow-on study.[6–8] BB:0–2 trial participants were eligible if women were nulliparous, aged 19 years or younger, were recruited at less than 25 weeks' gestation, living within the catchment area of a local FNP team, able to provide consent and speak English. Women expecting multiple births and those with a previous pregnancy ending in miscarriage, stillbirth or termination were eligible. Women planning to have their child adopted or to move beyond the FNP catchment area for more than 3 months were not eligible.[6–9] Trial participants had been randomly assigned to FNP or usual care (UC), with randomisation stratified by site and minimised by gestation (<16 weeks vs ≥16 weeks), smoking (yes vs no) and preferred language of data collection (English vs non-English) and weighted towards minimising the imbalance in trial groups with probability 0.8.

### Intervention and comparator (BB:0–2 trial)
The FNP programme provides up to a maximum of 64 home visits from specially trained family nurses during pregnancy until the child is 2 years old addressing personal and environmental health, life course development, maternal role, family and friends and access to health and social services. For the 697 women randomised and enrolled to FNP in BB:0–2 and then retained in the BB:2–6 cohort, the average number (SD) of valid visits reported by FNs as being received by FNP clients by programme phase was: pregnancy 9.74 (3.42), infancy 18.7 (5.97) and toddlerhood 13.28 (5.46) (online supplemental table S1). In the BB:0–2 trial, mean (SD) visit duration varied by delivery phase (pregnancy: 79.14 mins (13.78), infancy: 73.17 (11.61), toddlerhood: 74.75 (13.50) which exceeded the programme target of 60 min.[8] This compares to a duration of 75–90 min (undifferentiated across the three programme delivery phases) reported for the Elmira and Memphis trials.[10] All trial participants received usually provided health and social care services. Participants in the UC arm received these services alone.[6 8 11] The number (SD) of midwifery contacts were recorded in the BB:0–2 trial and were for FNP clients 10.68 (5.25) and for women in UC arm 10.69

(5.34).[11] The number (SD) of home visits from specialist public health nurses (SPHN) reported by trial participants by 18 months post partum was for FNP clients 4.70 (7.81) and for women in the UC arm 5.01 (5.5). The average number (SD) of clinic contacts with SPHNs was 0.70 (2.29) for women in the FNP arm and 6.31 (7.07) for women in the control arm.

## Procedure and data sources

The BB:0–2 trial consented 1618 eligible women between June 2009 and July 2010 and we concluded all BB:2–6 follow-up in May 2018. Maternal and child identifiers were sent to NHS Digital and the Department for Education (DfE) for matching with their databases. Maternal identifiers were also sent to the Department of Health and Social Care (DHSC) for matching with abortion statistics. Matching with NHS Digital and DHSC used a combination of NHS number, date of birth, postcode and sex. NHS Digital also provided a linked Hospital Episode Statistics (HES)—Office for National Statistics extract for mortality data. The process of matching for DfE records involved exact matching on first name and surname, date of birth and postcode. For social care data this was for both mother and child. For all other DfE datasets, this was just for the child. All matched data were sent to a third-party data safe haven (Secure Anonymised Information Linkage (SAIL) Databank) and linked by project identifiers to trial data and analysed via remote access.[12–14]

## Outcomes

The primary outcome was Child in Need (CIN) status recorded at any time by a local department of children's social care services (CSCS) (and sourced from the NPD). This includes children assessed as unlikely to achieve or maintain a reasonable level of health or development, or whose health and development is likely to be significantly or further impaired, without the provision of services; or is a child who is disabled. Secondary outcomes were additional formally reported measures of maltreatment (referral to CSCS, child being on a Child Protection Plan, CIN categorisation, Looked after status (mother, child)); Associated measures of maltreatment (recorded injuries and ingestions, non-attendance rates for hospital appointments); Maternal outcomes (subsequent pregnancies); Child health, developmental and educational outcomes (special educational needs, early educational attendance and assessments (Early Years Foundation Stage Profile (EYFSP), KS1)); and health resource use. Study assessment domains, outcomes and principal data providers and analysis are described in online supplemental table S2.

## Power calculation

Our sample size was fixed by the cohort available from BB:0–2. However, for CIN status, while UK data on rates are not specific to the age-range of interest, the rate per 10 000 general population aged 5–9 years is 4.6% for local authorities covering BB:0–2 trial sites.[15] The rate of CIN

status would be expected to be greater in the specific study sample. Therefore, we assumed a rate of 8%, hypothesised that FNP would reduce the presence of CIN in the first six years and assumed a difference of 4% would be important. To detect a difference of 4% (4% vs 8%) would require 602 children in each arm with 80% power and a two-sided 5% alpha level. With 1562 children available for follow-up (ie, excluding mandatory withdrawals) and follow-up through administrative records with 10% loss in tracking and linkage we would have 1405 participants, sufficient data to assess the primary outcome.

## Efforts to address sources of bias

All maltreatment and child developmental outcomes were independently state-verified (social worker) reports or teacher-assessments respectively, recorded in administrative service data and abstracted from national data providers. By using administrative data and established linking fields we were able to minimise bias due to loss to follow-up. An a priori statistical analysis plan described primary, secondary (including subgroup) and sensitivity analyses, and any additional exploratory analysis is clearly highlighted.

## Statistical analysis

Our analysis involved participants whose identifiers could be sent, linked and records released by the respective data providers. This excluded participants whose individual data could be used but could not be matched due to incorrect linking fields, other exclusions from health or education: for example, private or home schooling and National opt-outs. Imbalance between the BB:0–2 and eligible BB:2–6 participants was quantified using descriptive statistics in baseline demographics, clinical and questionnaire data. We conducted all analyses on an intention-to-treat basis with complete case population (with follow-up data). Therefore, participants remained in the BB:0–2 trial groups they were randomised to regardless of the intervention received. We used multilevel modelling to allow for clustering of effect within the 18 sites and family nurse level and both fitted as random effects. Where clustering was minimal at the family nurse level, results from the two-level model are presented. Parameter estimates are reported alongside a 95% CI and p value. We adjusted for variables used in randomisation (smoking status, gestational age, language). We defined loss to follow-up as a child death or adoption. We were able to determine both of these outcomes. We excluded children from analyses of binary outcomes where follow-up was incomplete and no event was observed. With time-to-event analyses, children were censored at these events.

Primary comparative analysis examined whether the firstborn child(ren) ('BB:0–2') had ever been referred to CSCS and classed as a CIN at any point between birth and 6 years of age (online supplemental appendix). We used logistic multilevel modelling to determine differences in the proportion defined as in need between the two trial arms. We present the resulting estimate as an OR. For

binary and categorical outcomes, comparative analysis used logistic and multinomial modelling, respectively, with results presented as OR. For continuous outcomes we used linear regression and present mean differences. For count data we used Poisson regression modelling; where event distribution displayed signs of over dispersion a negative binomial regression model was used (or zero-inflated model if appropriate). Results are presented as incidence rate ratios. Time to event data used Cox regression modelling presented with HR. All estimates are accompanied by a 95% confidence interval (CI). To reduce problems with multiple comparisons, some outcomes were only assessed descriptively (for example, we prioritised and tested overall scores for a Good Level of Development in the Early Years Profile, rather than the individual areas of learning). Similarly, for outcomes that we expected to reflect a small number of events or higher levels of missingness we also took a descriptive approach. Nevertheless, we present all a priori comparative analyses in the main tables and all other descriptive and exploratory analyses in online supplemental materials.

We received KS1 data in two waves from NPD. The first wave was for children assessed in the 2016/2017 academic year (received in May 2018). The second wave was for younger children, assessed in 2017/2018 (received in May 2019). Reading, maths and science were examined for the whole cohort over both academic years. However, the 2017/2018 writing assessment was changed with the consequence that judgements in 2018 are not directly comparable to judgements made using the previous interim frameworks (2016/2017). Therefore, we only assessed statistically writing using 2016/2017 data. We presented the data from 2017/2018 descriptively. Sensitivity analyses for primary and selected secondary outcomes (EYFSP, KS1, referral to CSCS) included adjusting for hypothesised confounders at baseline, dosage effects (ie, number of visits) using complier average causal effects modelling and subgroup analysis of potential effect moderators and mediators (maternal deprivation, adaptive functioning, NEET (not in education, employment or training) status at recruitment, maternal age at recruitment, child sex, maternal care status, duration of maternal care and domestic abuse self-reported at 24 months) as interaction terms in the main comparative models.

Results of all planned sensitivity analyses are presented in the appendix. An additional post hoc sensitivity analysis adjusted for the child's month of birth (categorised into quarter of birth) in the EYFSP and KS1 outcomes.

## Economic analysis

A cost–consequences analysis of FNP over the full follow-up period (BB:0–2 and BB:2–6) took a secondary healthcare (UK NHS) perspective. The principal data source was HES records (inpatient, outpatient, A&E), which were coded in Healthcare Resource Group Grouper and matched to appropriate NHS Reference costs. Maternal and child resource use were costed separately and valued in £ sterling. Only direct healthcare costs were analysed. Indirect costs such as lost wages or childcare costs were not available within the routinely collected data. Where data were absent in HES records it was assumed no resource was incurred. No primary care data were available beyond 18 months post partum. Costs were discounted back from year of event to baseline (2009/2010) at an annual rate of 3.5% as described by Central Government's guidance on appraisal and evaluation.[16]

## Data management

Small numbers were handled according to SAIL Databank rules where any cell counts under five were suppressed in reporting. Where abortion data are presented, we handled in accordance with the data sharing agreement with DHSC where counts less than 9 were suppressed. Data were analysed using Stata v16 and IBM SPSS V.23.0.

## Patient and public involvement

Activities involved four groups of young people (CASCADE Voices, Our Place, FNP graduates and ALPHA). These activities contributed to three aspects of the study: data-linkage methods (eg, to optimise acceptability of the proposed dissent model, use of data linkage and communication of these methods),[17] study outcomes (by seeking views on the importance of the study outcomes) and dissemination of findings (including best way to present results, method of dissemination).

## RESULTS

The study population following the BB:0–2 trial tracing and matching process is shown in figure 1. Five women were ineligible due to not meeting original trial entry criteria and 78 were mandatory withdrawals mainly due to fetal or infant death and adoption.[6] The 110 women who electively withdrew during the trial were given the opportunity to opt-out from further data usage, 16 of whom removed consent for any further contact. Therefore, we sent the details for these 94 women for tracing alongside the remaining 1452 mothers (a total of 1546 mother–child dyads) to update their contact details and thereby improve matching. One dyad was subsequently removed as deceased and the remaining 1545 contacted by letter, email or text of whom eight dissented from having their records linked. A total of 1537 mothers and 1547 children were sent to the DfE and NHS Digital for matching. These formed the population for the BB:2–6 study. Match rates for children were 98.3% (NHS Digital) and 97.4% (DfE).

Numbers matched to and returned for analysis differed by data source and population (online supplemental table S3). Few mothers, compared with children, were matched to any DfE data set (19% vs 97%, respectively) due to having ceased their education. These events were based mainly on CIN data. For selected maternal and child baseline variables and delivery of FNP assessed against programme fidelity goals, the BB:2–6 sample appears broadly representative of the original BB:0–2

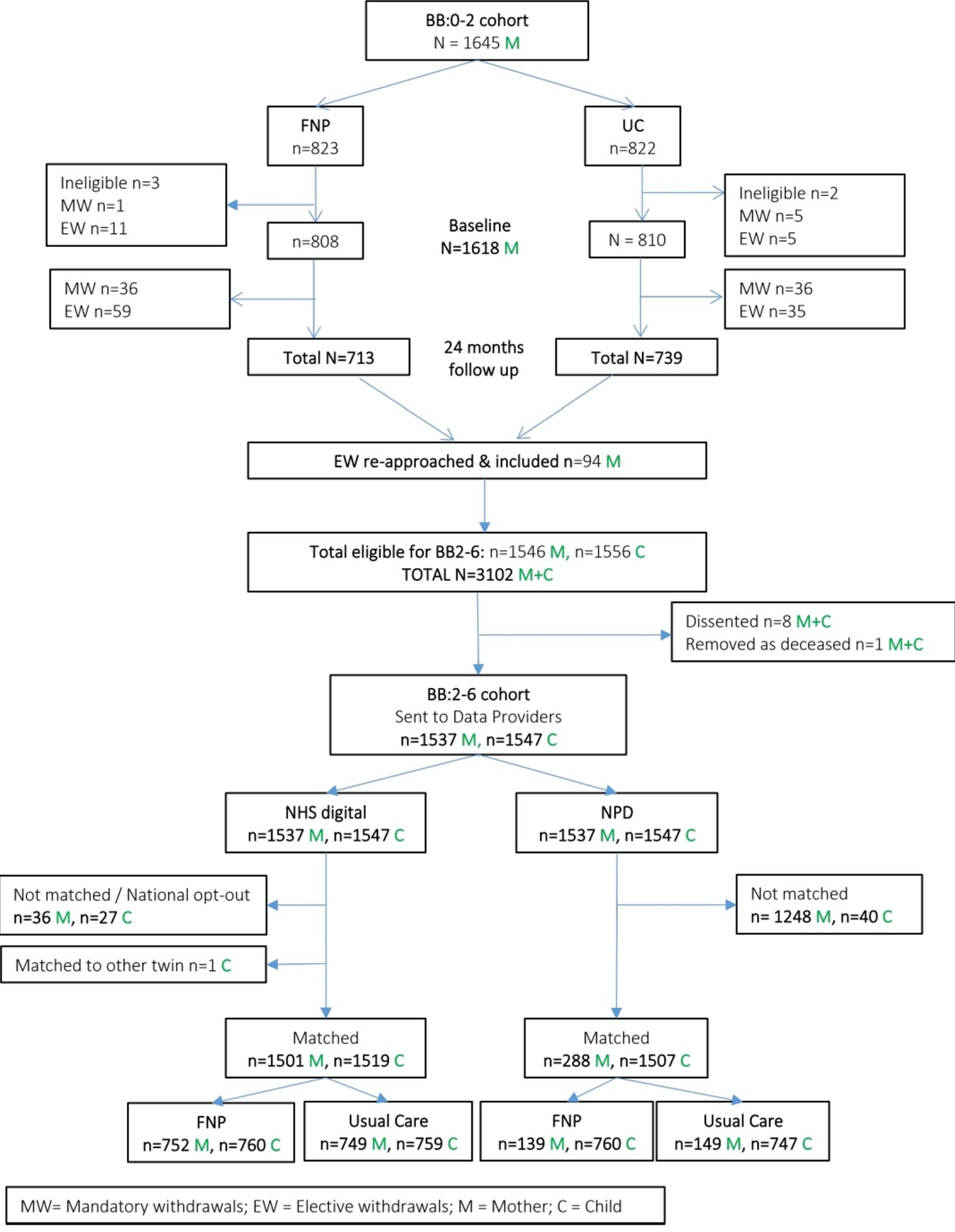

**Figure 1** Participant flow chart. Describing follow-up within the BB:0–2 trial and formation of the BB:2–6 study cohort. BB:0–2, Building blocks randomised controlled trial; BB:2–6, Building blocks cohort study; FNP, family nurse partnership (trial arm); NHS, National Health Service; NPD, National pupil database; UC, usual care (trial arm).

cohort (online supplemental tables S4,S5). Baseline characteristics of the BB:2–6 mothers and children were well balanced between trial arms (table 1).

We excluded referrals resulting in no further action (ie, children not classed as in need) for the definition of a CIN. Overall, 323 (21.5%) of children in the BB:2–6 cohort had at least one referral to CSCS which resulted as being classed as a CIN at any point between pregnancy (includes pre-birth CIN) and 6 years of age. No difference in rate of children in need was found between FNP (21.2%) and UC (21.7%) arms for the 1506 children in the analysis (adjusted OR=0.98, 95% CI=0.74, 1.31) (table 2). The parameter estimate was not affected when adjusting for the number of FNP visits received (efficacy per visit OR 1.00, 95% CI 0.99 to 1.004). The number of children referred to CSC and assessed as not requiring additional support were FNP (n=45) and UC (n=46). There were no between arm differences for referral to CSCS, the age at which the first referral was made, the age at which the child was classified as in need and the proportion of children with a protection plan. A small number of children were ever Looked After within the 6-year follow-up period, with no evidence of a difference between trial arms (table 2). However, the UC arm experienced on average two more months in care compared with the FNP arm. The total number of child deaths across the full follow-up period was less than ten and not further reported by study arm, consistent with the non-disclosive data management policy. Data informing the disability outcome were not available for the years requested. In addition, no difference was found when examining age at first referral or CIN by study arm.

No between arm differences were found in children not taken to at least one outpatient appointment (did not attend rate), emergency attendances or hospital admissions for an injury or an ingestion (table 2). Overall, 78% of mothers had a subsequent pregnancy (using inpatients, outpatients and abortions data) with no differences by arm (table 3). Similarly, just over two-thirds of the cohort had at least one registerable birth (using only admission data as a source) during the follow-up period with a comparable interbirth interval. Descriptives for non-tested maltreatment outcomes are shown in online supplemental table S6. Descriptives for the number of emergency room attendances and hospital admissions for injuries or ingestions are shown for each trial arm in online supplemental tables S7,S8. Exploratory analysis found no differences in length of hospital stay between trial arms following admission for an injury or ingestion for children aged under 1 year old or for children aged between 1 year old and under 2 years old (online supplemental table S8). Exploratory analyses of hospital admissions for injuries and ingestions by trial arm shown by age categories (Under 1 year, 1–2 years, 25 months to under 6 years) are shown in online supplemental table S9.

Evidence of positive programme impacts were found for some educational outcomes assessed at the end of the reception year and at KS1 by which time the child

**Table 1** Sociodemographic characteristics of mother and babies at baseline by trial arm

| | FNP N=766 | Usual care N=771 |
|---|---|---|
| Maternal age at recruitment (years) | 17.9 (17.0 to 18.8) | 17.9 (16.9 to 18.8) |
| Ethnicity | | |
| White background | 675 (88.1%) | 680 (88.2%) |
| Mixed background | 44 (5.7%) | 38 (4.9%) |
| Asian background | 15 (2.0%) | 10 (1.3%) |
| Black background | 29 (3.8%) | 40 (5.2%) |
| Chinese or other background | <5 | <5 |
| Relationship status with baby's father | | |
| Married | 6 (0.8%) | 10 (1.3%) |
| Separated | 72 (9.4%) | 78 (10.1%) |
| Closely involved/boyfriend | 582 (76.0%) | 586 (76.0%) |
| Just friends | 106 (13.8%) | 97 (12.6%) |
| NEET status* | | |
| Yes | 315/764 (41.1%) | 319/769 (41.4%) |
| No | 345/764 (45.0%) | 333/769 (43.2%) |
| IMD Overall Score† | 38.1 (24.6 to 52.6)‡ | 38.1 (25.5 to 51.6)§ |
| Generalised self-efficacy scale (score 10 to 40)¶ | 30.0 (28.0 to 33.0)** | 30.0 (27.0 to 32.0)†† |
| Adaptive functioning: | | |
| Difficulty in at least one basic skill | | |
| Yes | 213/765 (27.8%) | 184/770 (23.9%) |
| No | 552/765 (72.2%) | 586/770 (76.1%) |
| Had three or less life skills (out of 5) | | |
| Yes | 185/763 (24.2%) | 218/769 (28.3%) |
| No | 578/763 (75.8%) | 551/769 (71.7%) |
| At least one burden | | |
| Yes | 214/759 (28.2%) | 237/767 (30.9%) |
| No | 545/759 (71.8%) | 530/767 (69.1%) |
| Cigarette smoking participant self-reported | | |
| Ever smoked | | |
| Yes | 615 (80.3%) | 612 (79.4%) |
| No | 151 (19.7%) | 159 (20.6%) |
| Child characteristics | FNP N=773 | Usual care N=774 |
| Sex | | |
| Male | 381/773 (49.3%) | 406/773 (52.5%) |
| Female | 392/773 (50.7%) | 367/773 (47.5%) |
| Birth weight (grams) mean (SD) | 3223.81 (606.0)‡‡ | 3215.52 (555.56)§§ |
| NNU admission (direct or subsequent) | | |
| Yes | 76/716 (10.6%) | 66/749 (8.8%) |
| No | 640/716 (89.4%) | 683/749 (91.2%) |

Data are n (%), n/N (%), or median (25–75th centile) unless otherwise stated. <5 = numbers suppressed.
*Definition of NEET status: Not in education employment or training (applicable only to those whose age at end of previous academic year at time of baseline interview was >16).
†Higher IMD score indicated more deprivation. Mean IMD score for England in 2010 was 21.67 Wilkinson et al.[43]
‡N=760.
§N=765.
¶Higher score indicates higher level of self-efficacy.
**N=764.
††N=769.
‡‡N=724.
§§N=752.
FNP, family nurse partnership; IMD, Index of Multiple Deprivation; NEET, not in education, employment or training; NNU, neonatal unit.

**Table 2** Primary and secondary outcomes: CSCS outcomes, healthcare encounters for injuries/ingestions and out-patient non-attendance

| Outcome | FNP | Usual care (UC) | Adjusted* parameter estimate (95% CI) | P value | Absolute risk difference (FNP-UC) (95% CI) |
|---|---|---|---|---|---|
| **Child protection outcomes** | **N=760** | **N=746** | | | |
| Child in need (CIN) status at any time during the follow–up period | | | | | |
| No | 599 (78.8%) | 584 (78.3%) | Reference | | |
| Yes | 161 (21.2%) | 162 (21.7%) | 0.98† (0.74 to 1.31) | 0.902 | −0.5% (−0.5 to 0.4%) |
| Age at first CIN referral (days) | | | | | |
| Median (25–75th centile) | 1602.5 (1153.0 to 1978.75) | 1649.5 (1130.75 to 1980.25) | 0.98‡ (0.79 to 1.23) | 0.875 | |
| Missing | 1 | 0 | | | |
| Unique no of CIN referrals per child§ | 1 (1 to 2) | 1 (1 to 2) | 0.99¶¶ (0.82 to 1.19) | 0.890 | |
| CIN duration (days) | 210 (70.5 to 571.25) | 216 (72.75 to 503.25) | 1.20‡ (0.90 to 1.59) | 0.216 | |
| CIN categorisation (primary need) of first case | **N=161** | **N=162** | | | |
| Abuse or neglect | 92 (57.1%) | 102 (63.0%) | Reference | | |
| Family dysfunction | 29 (18.0%) | 34 (21.0%) | 0.95¶ (0.54 to 1.69) | 0.868 | |
| Family in acute stress | 13 (8.1%) | 10 (6.2%) | 1.47¶ (0.61 to 3.54) | 0.389 | |
| Low income, absent parenting, socially unacceptable behaviour, not stated | 13 (8.1%) | 6 (3.7%) | 2.38§ (0.86 to 6.57) | 0.093 | |
| Child/parent disability | 7 (4.3%) | 5 (3.1%) | 1.47¶ (0.45 to 4.84) | 0.525 | |
| Cases other than CIN | 7 (4.3%) | 5 (3.1%) | 1.64¶ (0.50 to 5.37) | 0.417 | |
| Referred to children's social care services (CSCS) | | | | | |
| No | 554 (72.9%) | 538 (72.1%) | Reference | | |
| Yes | 206 (27.1%) | 208 (27.9%) | 0.97† (0.74 to 1.28) | 0.829 | −0.8% (−5.3 to 3.7%) |
| Age at first referral (days) Median (25–75th centiles) | 1541.5 (1272.75 to 1975.75) | 1631.0 (1190.5 to 1984.75) | 0.96‡ (0.79 to 1.17) | 0.694 | |
| Child protection plan | | | | | |
| No | 708 (93.2%) | 697 (93.4%) | Reference | | |
| Yes | 52 (6.8%) | 49 (6.6%) | 1.04† (0.69 to 1.57) | 0.846 | 0.2% (−0.2 to 0.3%) |
| Category of child protection plan | **n=52** | **n=49** | | | |
| Neglect | 17 (32.7%) | 22 (44.9%) | Reference | | |
| Emotional | 22 (42.3%) | 12 (24.5%) | 2.77¶¶ (1.02 to 7.56) | 0.046 | |
| Physical | 6 (11.5%) | 6 (12.2%) | 1.25¶¶ (0.32 to 4.88) | 0.743 | |
| Sexual and multiple** | 7 (13.5%) | 9 (18.4%) | 1.13¶¶ (0.33 to 3.86) | 0.848 | |
| Looked after status†† | | | | | |
| No | 735 (96.7%) | 719 (96.4%) | Reference | | |
| Yes | 25 (3.3%) | 27 (3.6%) | 0.90† (0.52 to 1.57) | 0.712 | −0.3% (−0.2 to 0.2%) |
| Child looked after period of care (months) | 10.0 (4.5 to 37.5) | 12.0 (6.0 to 33.0) | 0.75§ (0.65 to 0.86) | <0.001 | |

Continued

**Table 2** Continued

| Outcome | FNP | Usual care (UC) | Adjusted* parameter estimate (95% CI) | P value | Absolute risk difference (FNP-UC) (95% CI) |
|---|---|---|---|---|---|
| Injuries and ingestions | N=760 | N=759 | | | |
| Emergency attendance§§ | | | | | |
| None | 317 (41.7%) | 344 (45.3%) | Reference | | |
| At least one | 443 (58.3%) | 415 (54.7%) | 1.17† (0.95 to 1.45) | 0.149 | 3.6% (−1.4 to 8.6%) |
| No of attendances per child | 2 (1 to 2) | 2 (1 to 3) | 1.09¶¶ (0.93 to 1.28) | 0.281 | |
| Admission to hospital | | | | | |
| None | 671 (88.3%) | 660 (87.0%) | Reference | | |
| At least one | 89 (11.7%) | 99 (13.0%) | 0.87† (0.63 to 1.20) | 0.407 | −1.3% (−4.7 to 2.0%) |
| Unique admissions | 109 | 119 | | | |
| Ratio of admissions to children | 1.22 | 1.20 | | | |
| No of admissions per child, median (25–75th centile) | 1 (1 to 1) | 1 (1 to 1) | 0.93¶¶ (0.67 to 1.29) | 0.663 | |
| Length of stay (days)‡‡ | N=106 | N=119 | | | |
| Median (25–75th centile) days | 0.5 (0.5 to 1.0) | 0.5 (0.5 to 1.0) | | | |
| Hospital attendance and/or admission | | | | | |
| None | 306 (40.3%) | 324 (42.7%) | Reference | | |
| At least one | 454 (59.7%) | 435 (57.3%) | 1.11† (0.89 to 1.37) | 0.351 | 2.4% (−2.5 to 7.4%) |
| DNA outpatient appointment | N=580 | N=577 | | | |
| Attended all appointments | 290 (50.0%) | 289 (50.1%) | Reference | | |
| DNA at least once | 290 (50.0%) | 288 (49.9%) | 1.00† (0.79 to 1.26) | 0.997 | 0.09% (−5.6 to 5.8%) |

Data are n (%), n/N (%) or median (25–75th centile) unless otherwise stated.
*FNP compared with usual care. Analysis adjusted for stratification (site), minimisation variables (gestational age, smoking status at recruitment, and first or preferred language).
†OR from logistic model.
‡HR from Cox model.
§A unique referral is counted as a distinct referral date per child.
¶Relative risk ratio from multinomial model.
**Multiple indicates when more than one category of abuse is relevant to the child's current plan.
††We reflect here the terminology used at the time of the trial but note the more contemporary 'care experienced' as a more familiar and accepted term.
‡‡0.5 days indicates an admission and discharge on the same day (could be up to 1 day in hospital). Data available from NHS Digital which show an admission and discharge 1 day apart may not equate to one full day in hospital (ie, the child may have been in hospital for only a few hours if the admission spanned midnight). Source: National Pupil Database, The Department for Education.
§§Using diagnosis in any position.
¶¶Incidence rate ratio from Poisson model.
FNP, family nurse partnership; NHS, National Health Service.

**Table 3** Secondary outcomes: subsequent pregnancies and registerable birth

| | FNP | Usual care | Adjusted* parameter estimate (95% CI) | P value | Absolute risk difference (FNP-usual care) (95% CI) |
|---|---|---|---|---|---|
| Subsequent pregnancy | N=753 | N=753 | | | |
| No | 163 (21.7%) | 163 (21.7%) | Reference | | |
| Yes | 590 (78.4%) | 590 (78.4%) | 1.00† (0.79 to 1.28) | 0.984 | 0.0% (−4.2% to 4.2%) |
| Subsequent registerable birth | N=752 | N=749 | | | |
| No | 276 (36.7%) | 266 (35.5%) | Reference | | |
| Yes | 476 (63.3%) | 483 (64.5%) | 0.95† (0.77 to 1.18) | 0.662 | −1.2% (−6.0% to 3.7%) |
| One birth | 326 (68.5) | 331 (68.5) | 0.95‡ (0.76 to 1.19) | 0.655 | |
| Two | 124 (26.1) | 121 (25.1) | 0.99‡ (0.73 to 1.34) | 0.955 | |
| Three births or more | 26 (5.5) | 31 (6.4) | 0.80‡ (0.46 to 1.39) | 0.435 | |
| Interbirth interval between first and second child (days) Median (25–75th centiles) | 1027 (590 to 1506.75) | 1065 (665 to 1538) | 0.99§ (0.88 to 1.13) | 0.938 | |

Data are n (%), n/N (%) or median (25–75th centile) unless otherwise stated.
*FNP compared with usual care. Analysis adjusted for stratification (site), minimisation variables (gestational age, smoking status at recruitment, and first or preferred language).
†OR from logistic model.
‡Relative risk ratio from multinomial model.
§HR from Cox model.
FNP, family nurse partnership.

would be 7 years old (table 4). Children in the FNP arm were more likely to reach a good level of development across all five areas of learning by the end of the reception year (58.0%) than children in the UC arm (52.2%) and also to achieve a good level of development in all 17 early learning goals (FNP: 55.5%, UC: 50.1%). The small advantage for children of nurse-visited mothers was consistent across all five areas of learning although not tested statistically (online supplemental table S10).

For KS1 reading, writing, maths and science assessments there was no evidence of a difference between arms in the proportion of children reaching at least the expected standard nor specifically in those working at the expected level or at a greater depth of knowledge (table 4). What is observed from the writing assessment is the increase in the rates of children reaching at least the expected standard between the two academic years (Overall 45.7% in 2016/2017% to 66.5% in 2017/2018), reflective of all assessments across the two academic years in each arm (online supplemental table S11). This is mainly explained by the distribution of births for the children for each academic year (online supplemental figure S1). Children with KS1 results for the academic year 2016/2017 were born between October 2009 and August 2010. However, there was a skew towards summer-born children (ie, 42% were born between June and August 2010). Children with KS1 results for the academic year 2017/2018 were born between September 2010 and February 2011. Nationally, there is a relationship between rates of children reaching the expected standard and month of birth (eg, there is a 17% difference in reading for children born in August and September).[18] Therefore, the 2017/2018 cohort of children are not fully representative of the whole academic year and are biased towards children who are more likely to achieve. This indicates that the rates of children reaching the expected standard varies by their month of birth and is important as a moderator of programme effect. Importantly, the original trial allocation ensured study arms remained balanced with regards month of birth and the intervention had no impact on total weeks gestation.[6] As a sensitivity analysis, the main analyses were adjusted for the child's month of birth (categorised into quarter of birth) and evidence of a between arm difference was found in reading assessments (online supplemental table S12). In addition, the Early Years assessments were adjusted for child's month of birth, which strengthened the association found in table 4.

Details of all planned subgroup analyses and further exploratory analyses are shown in the appendix (online supplemental tables S13–S18). The latter included exploration of potential surveillance bias for all children assessed as a CIN by age 4 years old. This explored whether children in the FNP arm may have been referred and assessed as in need at lower levels of concern than children in the UC arm based on maternal baseline characteristics (online supplemental table S18). Mothers in the UC arm were more likely to have not been in education, employment or training than mothers in the FNP arm but there were no other differences at baseline. The two groups of children assessed as in need were also no different on subsequent measures of school readiness.

## Cost

There were negligible resource use and cost differences for both mothers (p=0.393) and children (p=0.865) between study arms (online supplemental table S19). The adjusted incremental costs of programme delivery

**Table 4**  Secondary outcomes: child health, developmental and educational

| Outcome | FNP | Usual care | Adjusted* OR (95% CI) | P value | Absolute risk difference (FNP-usual care) (95% CI) |
|---|---|---|---|---|---|
| Special educational needs (SEN) provision† | N=759 | N=747 | | | |
| No | 540 (71.1%) | 502 (67.2%) | Reference | | |
| Yes | 219 (28.9%) | 245 (32.8%) | 0.83 (0.67 to 1.03) | 0.097 | −3.9% (−8.6% to 0.7%) |
| Early educational attendance | N=759 | N=747 | | | |
| Attending an Ofsted registered private, voluntary and independent establishment up to the age of 4 years | 334 (43.9%) | 308 (41.2%) | 1.13 (0.91 to 1.40) | 0.281 | 2.7% (−2.3% to 7.7%) |
| School attendance | N=754 | N=740 | | | |
| Overall absence | | | | | |
| No absences | 14 (1.9%) | 14 (1.9%) | Ref | | |
| At least one absence | 740 (98.1%) | 726 (98.1%) | 1.00 (0.47 to 2.12) | 0.998 | 0.0% (−1.4% to 1.5%) |
| Overall authorised absence | | | | | |
| No absences | 26 (3.4%) | 26 (3.5%) | Ref | | |
| At least one absence | 728 (96.6%) | 714 (96.5%) | 1.01 (0.58 to 1.75) | 0.984 | 0.1% (−1.8% to 2.8%) |
| Overall unauthorised absence | | | | | |
| No absences | 256 (34.0%) | 245 (33.1%) | Ref | | |
| At least one absence | 498 (66.0%) | 495 (66.9%) | 0.95 (0.76 to 1.18) | 0.620 | −0.8% (−5.6% to 3.9%) |
| Early years assessment | N=743 | N=728 | | | |
| Achieving good level of development (GLD)‡ | | | | | |
| Achieving GLD in all five areas of learning§ | 431 (58.0%) | 380 (52.2%) | 1.26 (1.03 to 1.55) | 0.026 | 5.8% (0.7% to 10.9%) |
| Achieving GLD in all 17 early learning goals | 412 (55.5%) | 365 (50.1%) | 1.24 (1.01 to 1.52) | 0.043 | 5.3% (0.2% to 10.4%) |
| Total point score§ Mean (SD) | 32.22 (7.25) | 31.59 (7.62) | 0.65§ (−0.11 to 1.41) | 0.094 | |
| Key stage 1 assessments | N=740 | N=732 | | | |
| Reading | | | | | |
| Lower than expected | 257 (34.7%) | 289 (39.5%) | Reference | | |
| Reaching at least the expected standard¶ | 483 (65.3%) | 443 (60.5%) | 1.23 (0.99 to 1.53) | 0.051 | 4.8 (−0.2% to 9.7%) |
| Expected standard | 371 (50.1%) | 337 (46.0%) | 1.24 (0.99 to 1.56) | 0.056 | 4.1 (−1.0% to 9.2%) |
| Higher standard | 112 (15.1%) | 106 (14.5%) | 1.20 (0.88 to 1.65) | 0.250 | 0.7 (−3.0% to 4.3%) |
| Maths | | | | | |
| Lower than expected | 281 (38.0%) | 283 (38.7%) | Reference | | |
| Reaching at least the expected standard¶ | 459 (62.0%) | 449 (61.3%) | 1.04 (0.84 to 1.28) | 0.731 | 0.7 (−4.3% to 5.6%) |
| Expected standard | 392 (53.0%) | 376 (51.4%) | 1.06 (0.85 to 1.32) | 0.611 | 1.6 (−3.5% to 6.7%) |
| Higher standard | 67 (9.1%) | 73 (10.0%) | 0.93 (0.64 to 1.35) | 0.711 | −0.9 (−3.9% to 2.1%) |
| Science | | | | | |
| Lower than expected | 203 (27.4%) | 219 (29.9%) | Reference | | |
| Reaching at least the expected standard** | 537 (72.6%) | 513 (70.1%) | 1.14 (0.91 to 1.43) | 0.254 | 2.5 (−2.1% to 7.1%) |
| Writing academic year 2016/2017 | N=498 | N=487 | | | |
| Lower than expected | 257 (51.6%) | 278 (57.1%) | Reference | | |
| Reaching at least the expected standard¶ | 241 (48.4%) | 209 (42.9%) | 1.24 (0.97 to 1.60) | 0.090 | 5.5 (−0.7% to 11.6%) |
| Expected standard | 218 (43.8%) | 182 (37.4%) | 1.29 (1.00 to 1.68) | 0.054 | 6.4 (0.3% to 12.5%) |
| Higher standard | 23 (4.6%) | 27 (5.5%) | 0.92 (0.51 to 1.64) | 0.769 | −0.9 (−3.8 to 1.9%) |

Data are n (%), n/N (%) or median (25–75th centile) unless otherwise stated.

*FNP compared with usual care. Analysis adjusted for stratification (site), minimisation variables (gestational age, smoking status at recruitment, and first or preferred language).

†A child with SEN provision is recorded using the following codes (A=School Action, p=School Action Plus, E=Education, Health and Care Plan, S=Statement, K=SEN support) and the N code was used to indicate No SEN support in any of the following datasets between 2013 and 2017. Pupil-level annual school census (PLASC) (Autumn, Spring and Summer terms), alternative provision (AP), pupil referral unit (PRU) Census and Early Years Census (EYC). If for any of the years no response was recorded then the assumption was that the child was not included in the denominator as either not present on Census day or not in school.

‡Children achieving a GLD are those achieving at least the expected level within the prime and specific areas of learning; §Total point score ranges from 17 to 51 with a higher score indicating a better level of development.

§Mean difference from linear model.

¶Working at the expected standard and at a greater depth within the expected standard.

**Working at a greater depth within the expected standard is not applicable in science. Source: National Pupil Database, The Department for Education.

FNP, family nurse partnership.

per women in BB:0–2 (£1,811) remain the key observed cost difference between study arms.

## DISCUSSION
### Principal findings

In this data linkage study, we found that adding FNP to usually provided health and social care provided no additional benefit for maltreatment outcomes including the numbers of children referred to CSCS, registered as in need of additional support, given a child protection plan, entering care, attending an emergency department or being admitted for an injury or ingestion, or duration of stay if admitted. While there was no difference between study arms in the small number of children in care, the UC arm experienced on average two more months in care compared with the FNP arm. The total number of child deaths found was less than ten and therefore disclosive and cannot be reported. High rates of children ever registered as in need in both study arms indicates substantial need among families. We found the programme led to more children reaching a good level of development at the end of school reception year and, when taking month of birth into account, improvements in reading at KS1. Writing scores improved as a result of FNP for boys, for children of younger mothers and for children of mothers who were NEET when recruited to the trial. We found no other difference between families who received FNP and those who did not. With no differences in secondary health resource use and costs for FNP, the programme can be considered cost-neutral compared with UC in the medium term (2–6 years).

### Strengths and weaknesses of the study

The prioritisation of outcomes in our original trial attracted some debate.[19–21] However, the trial and the current follow-on study are both comprehensively reported, and preceded by a priori analysis plans providing transparency and guarding against undue post hoc interpretation.[7–9] Although a primary outcome of child being in need of additional support was incorporated into the study design, we adopted a multimethod multisource approach to look at the range of maltreatment outcomes in drawing our conclusions. The consistent pattern of results within the two principal outcome domains provides greater confidence in what we can conclude. While the study benefited from random allocation in the BB:0–2 trial, when assessing educational outcomes we additionally adjusted for month of birth. This was because at a national level, month of birth is strongly associated with educational outcomes and the distribution of participants recruited to the trial varied by month.

The available maltreatment data provides an essential overview of family experience as recorded in official records but may lack some depth which could have allowed a more nuanced understanding for children referred. The educational data are teacher-reported statutory tests but similarly provide less detailed insight into child functioning which research instruments may have provided. The lack of detailed social care and any primary care resource use data limited the perspective of the economic analysis. However, using administrative data has enabled comprehensive cohort follow-up, with few cases lost for analysis preserving the benefits of the original trial randomisation. The data included are also objectively recorded rather than maternally self-reported, allow for easy comparison nationally and over time. Our approach has gathered more data at less cost than comparable prospective data collection and minimised family burden.

In our previous trial report of outcomes at 24 months post partum, there were higher rates of safeguarding events recorded in GP notes for children in the FNP arm (n=64/469, 13.6%), compared with the UC arm (n=38/476, 8.0%).[6] Additionally, more mothers in the FNP arm reported their child had ever been referred to social services (n=119/580, 20.5%) compared with mothers in the UC arm (n=91/541, 16.8%). These finding can be indicative of surveillance bias or more accurate reporting by mothers in the FNP arm respectively. In the current study a tendency for children to be referred at lower thresholds of concern or closer involvement by attending FNP nurses may have been evidenced through a greater proportion of children referred to social services being assessed as requiring no additional support. However, we did not observe this in our cohort.

With no absolute external criterion for maltreatment, a reliance solely on formal child protection reports may be limiting. In the US Elmira trial, a review of 42 children (intervention arm n=13, control arm n=29) referred to child protective services (CPS - the term used in the US setting) between birth and their fourth year of life found few differences in the nature of maltreatment recorded. However, measures of the quality of home environment, parenting, child intellectual functioning and use of healthcare services between children's 25th and 50th months of life favoured families in the intervention arm.[22] It has been suggested that both parental and professional behaviour may lead to programme enrolled CPS reported children being at lower risk of harm than their counterparts in receipt of UC alone. Thresholds for referral to social services may differ between specialist home-visiting nurses with high contact time, more in-depth relationship with clients and different approaches to risk and safety management and other health, social and educational care professionals.[23] For children referred to CSCS and assessed as a CIN, we explored whether the trial arms differed on characteristics that may suggest a lower threshold of concern for those in the FNP arm. Our comparative analysis was limited principally to baseline maternal characteristics to reduce the risk of interpreting outcomes subsequent to the CIN assessment as evidence of contemporary child concerns. This exploratory analysis found a higher rate of educational, employment and training inactivity at trial baseline among mothers in the

UC arm compared with the FNP trial arm. While no other differences were found, this is consistent to the findings in the Elmira trial.[22] CPS referrals for children in receipt of FNP at lower levels of apparent concern would make it harder to compare on such outcomes, particularly during the programme delivery period (ie, up to the child's second birthday). However, subsequent progress of children through the child protection system and later occurring outcomes (eg, children being taken into care) may be less liable to such influences but in our sample such numbers are small.

Verified outcomes documented in administrative records contrast with maternal subjective self-reports for which there may be no potential external criterion (ie, a child being referred to CPS is a verifiable event, whereas maternal well-being is not). Nevertheless, the presentation or referral of families and their assessment against legally defined statuses such as being a CIN requires professional judgement. It has been suggested that surveillance bias is not likely to be problematic for assessing maltreatment using verified service data (ie, by inflating the numbers of children being reported when in receipt of home-visiting).[24 25] As others have found this could be higher when clients are actively engaged with services and as noted above, the potential for such bias remains.[26]

It has been proposed that other documented evidence such as hospital admission due to injury and ingestions and in particular duration of subsequent stay may provide a better indicator of maltreatment severity. In the follow-up of the US Memphis trial cohort at 2 years of the child's life, children in the intervention arm admitted following an injury or ingestion spent fewer days in hospital than their counterparts in the control arm. In the US Elmira trial, no difference between trial arms were found in the number of days hospitalised during the 25–50 months period of follow-up after adjustments were made for one outlier (in the intervention arm) assessed as being not relevant to parental caregiving.[27] We explored length of hospital stay following admission for an injury and ingestion both overall and for time periods representing periods of increasing of child mobility (ie, under 1 year of age, 1 to under 2 years of age and 25 months to under 6 years of age). These were similar in each case but factors other than severity of clinical need may influence how long children remain in hospital. Wide differences exist between countries for hospital stay duration following injuries, ingestions and other consequences of external causation (when assessed across adult and paediatric populations).[28] One observational UK study of predictors of length of stay among paediatric inpatients (all causes) shows the contribution of health service factors in addition to clinical and social factors.[29] Interestingly, a large US study among paediatric inpatients found that non-accidental trauma compared with accidental trauma was a strong predictor of having a prolonged hospital stay even after adjusting for injury severity.[30] Therefore, length of hospital stay may be indicative of maltreatment,

severity of clinical presentation, health service factors and possibly social factors. Further clarifying the factors that independently drive length of hospital stay and the validity of this outcome as a measure of maltreatment will be valuable.

## Potential sources of bias

The comparisons being made in this study are principally between women in receipt of visits from family nurses, in addition to usually provided health and social care and women receiving usually provided care only. In our previous full trial report, we summarised participant-reported contacts with a wide range of health and social care services up to 18 months post partum. In addition, the within-trial economic analysis involved imputing values for all resource domains to cover the final 6 months of the trial period (ie, up to 24 months post partum) and which assumed equivalent resource use at 24 months as reported at 18 months.[8] This included resource use (ie, contacts) related to SPHN. The full trial report includes descriptions of both reported and imputed values; however, it should be noted that the concurrently published journal article only included the higher values imputed for the purpose of the economic analysis.[6] As the schedule for delivery of the universal elements of the healthy child Programme (HCP) specifically by SPHNs indicates a decreasing frequency of contact over time (and none from 1 year until by 2–2.5 years) this is likely to be a conservative (overestimate) of actual provision.[31]

As an open-label study, it is possible that service professionals may have attempted to compensate for the lack of specialist support from FNP for participants in the UC arm. However, it should be noted that enhanced care would have been expected as part of usual practice for teenage mothers, for example, through specialist teenage pregnancy midwives who were employed at 14 of the 18 trial sites, with lower caseloads and an emphasis on home-based antenatal care, promotion of healthy relationships and positive parenting. For example, the specification for children's public health services (from pregnancy to age 5) indicates a schedule of universal elements for the Healthy Child Programme, including both universal and progressive elements where young mothers are not under the care of family nurses.[31] The number of midwifery contacts were recorded in the trial and found to be equivalent between FNP and UC arms.[11] The number of home visits from SPHNs reported by trial participants by 18 months was also equivalent between study arms. As would have been expected, the number of clinic contacts with SPHNs was greater for women in the control arm than in the FNP arm. In contrast, women allocated to the intervention arm would have received an average of just over 40 home visits from a family nurse before graduating from the programme. Overall, this suggests a common core of UC provided across both trial arms, a higher level of SPHN input to the UC arm reflecting in part the responsibility of delivering the Healthy Child Programme (which was formally allocated to FNP nurses

rather than SPHNs for their clients). The capacity of SPHNs to provide additional support will also be limited by their caseload. The maximum caseload currently recommended by the Institute of Health Visiting is one practitioner per 250 children. However, in 2015, 28% of surveyed practitioners in England reported caseloads in excess of 400 children and 12% reported between 500 and 1000+ families.[32] Providing additional support only to mothers with apparent need rather than all women on each SPHN's caseload may be considered at least more feasible. If so, and if this actually approached an optimal level of support driven by women's needs as per the progressive universal model then our trial comparison of FNP would still be against best UC. There are limitations in the data captured on routine health service usage (see relevant excerpts of case report form data for postpartum data collection in Appendix XII). For example, SPHN contact content and duration were not captured. This was due to pragmatic and ethical considerations (eg, to not increasingly burden participants for what were already lengthy telephone data collection schedules). Similarly, reported contacts between women in the trial intervention arm and SPHNs would have been worthy of further investigation. It's possible that some reports could in fact have been related to family nurses rather than SPHNs, although perhaps less likely for women engaged in a constructive on-going relationship with a named family nurse. Furthermore, women in the intervention arm who either never engaged with FNP or may have dropped scheduled contact with their family nurse would be expected to have had ongoing contact with a SPHN.

The potential for the open-label nature of the study to introduce bias through the actions of SPHNs is an important question and can be viewed through the lens of contemporary behavioural change frameworks such as COM-B.[33] This approach looks at three essential conditions required for a Behaviour change to occur (eg, in this instance a SPHN acting to provide enhanced support to a mother)—Capability, Opportunity and Motivation. First, do SPHNs have the skills, training, model of delivery and supportive management structure (ie, Capability) to be provide an enhanced service? Many SPHNs would have had similar professional backgrounds and skills to their FNP colleagues, and some expressed similar practice aspirations to those they perceived among family nurses. In contrast, most FNP nurses would have been selectively recruited from existing SPHNs and received additional programme training, supervision and a distinct programme delivery model. While the capabilities of SPHNs and FNP nurses certainly overlapped, they were unlikely to be equivalent at the time of the trial. Second, as noted above, the opportunity for SPHNs to deliver additional support would have been limited by their caseload. This was evidenced by the testimony of SPHNs in the trial's process evaluation and reported caseloads of up to 600 women at two trial sites.[8] The reduction to an individual SPHN's own caseload associated with an FNP nurse enrolling up to her caseload maximum of 25

clients would appear to offer limited scope to substantially increase the opportunity to then enhance provision. Finally, would SPHNs be motivated to enhance service provision either in general (eg, to all clients who may be seen as likely to benefit from additional support) or specifically to participants allocated to the trial's control arm? In our trial's process evaluation, several SPHNs reported a lack of awareness about whether their clients were in the trial and in some cases a misperception that they should not know either. SPHNs with the potential to provide additional care and to do so specifically for women they knew to be enrolled in the trial and allocated to the control arm (and not say to other women with similar needs unwilling to participate in the trial or who were not identified to do so) would require considerable motivation. To do so would also represent a wilful attempt to undermine the requirements of a trial supported by their own departments. For this to occur consistently for a large number of SPHNs both within and across the 18 trial sites at a level that may start to impact on outcomes seems less likely. Given the 'fire-fighting' described by SPHNs in delivering care as usual to their large caseloads, the additional motivation to target additional support to women known to be in the trial seems more a theoretical rather than a practical risk.

### Comparison with other studies

The strongest evidence for the programme preventing maltreatment remains from the original US Elmira trial. While the programme reduced the number of substantiated reports of abuse and neglect in the first 15 years of the child's life, the advantage emerged after age 4 years.[4] We found differences in duration spent in care for looked after children, but the numbers involved are small and across the range of maltreatment outcomes the pattern is of no overall group difference. The programme was no more effective in tested subgroups. It is worth recognising differences in the population of clients in receipt of the programme across the different study populations, their social circumstances (which also may have changed over time) and how that may determine potential to benefit from FNP.[11] We have not directly explored differences in social care systems between England and the USA but these may vary the underlying risk of maltreatment and its likelihood of detection.[34]

School entry reading and maths is of important predictive value for later school achievement.[35] Early childhood educational investment programmes typically show a modest impact on cognitive and achievement scores (eg, weighted average of 0.21 SD) with some decline in observed impact being attributed to general improvements in both school and home environments.[36 37] The question of comparator is therefore of particular relevance. In this study, a range of state and third sector funded services (including midwifery and health visiting) would have been available as UC, possibly limiting the incremental benefit possible from the FNP programme.[11] Nevertheless, as noted above women allocated to FNP

received substantial input from family nurses, although fewer than possible according to the maximum fidelity targets at the time of the trial.

Home visiting interventions, which aim to impact on a range of outcomes, have shown similarly modest effects on child cognitive outcomes (eg, mean effect size 0.25, 95% CI 0.11, 0.38) with considerable variability due to programme composition (eg, greater effects with programmes addressing parental responsiveness, sensitivity to care and nurturing and rehearsal or role-playing).[38] Our BB:0–2 study found benefits at age 2 years for children in the FNP arm for both language and developmental outcomes but these outcomes were maternally reported and potentially subject to reporting bias in the open trial.[9] This new medium-term evidence for programme impact on developmental outcomes adds to that picture, although the size of effect remains small. Similar medium-term programme impact (to age 6 years old) on developmental outcomes (eg, intellectual functioning and receptive language scores) was reported from the Memphis trial.[39] A reanalysis of those data found cognitive benefits at age 6 years attributed to both programme induced improvements in maternal traits and family life investments at age 2.[40]

### Unanswered questions and future research

Given small but positive findings for developmental outcomes, tracking the cohort through to subsequent school years is warranted. As social care data are equally accessible via the NPD there is value in observing the trajectory for maltreatment outcomes given that the principal maltreatment findings from the US Elmira trial are located in the 15-year follow-up. Linkage to data in other sectors such as welfare would provide a greater understanding of broader impact for families. Such work should be driven by the programme's logic model, existing evidence for long-term programme impact, informed by programme evolution in England and supplemented by work to understand what may explain differing programme benefits across national settings.[4 41]

### Conclusions

There are no evident programme benefits for maltreatment outcomes by age six but the FNP generates higher rates of child attainment at the end of the reception year and at KS1. The adapted programme remains locally commissioned and delivered in England.[41 42] Local needs and priorities may determine the weight attached to these different sets of outcomes.

**Author affiliations**
[1]Centre for Trials Research, Cardiff University, Cardiff, UK
[2]DECIPHer, School of Social Sciences, Cardiff University, Cardiff, UK
[3]Swansea Centre for Health Economics, Swansea University, Swansea, UK
[4]Faculty of Life Sciences and Education, University of South Wales, Pontypridd, UK
[5]School of Healthcare Sciences, Cardiff University, Cardiff, UK
[6]School of Social Sciences, Cardiff University, Cardiff, UK

**Acknowledgements** The Centre for Trials Research receives funding from Health and Care Research Wales and Cancer Research UK. The work was undertaken with the support of The Centre for the Development and Evaluation of Complex Interventions for Public Health Improvement (DECIPHer), a UKCRC Public Health Research Centre of Excellence. Joint funding (MR/KO232331/1) from the British Heart Foundation, Cancer Research UK, Economic and Social Research Council, Medical Research Council, the Welsh Government and the Wellcome Trust, under the auspices of the UK Clinical Research Collaboration, is gratefully acknowledged.

**Contributors** MR was the chief investigator and acts as guarantor for the study. MR, FVL-W, RC-J, LA, SC, DF, KH, JK, GM, EO-J, RDP, JSa, JSe and TS contributed to study design. FVL-W was the study manager. RC-J was the lead statistician. DF supervised the health economics. MR, FVL-W, RC-J, LA, SC, DF, KH, JK, GM, EO-J, RDP, JSa, JSe and TS were responsible for study management. GM and LA were the data managers. JK and JS led for public involvement. GM and LA cleaned the data. RC-J designed and conducted the analysis. RDP conducted the economic analysis. MR, JSa, RC-J, SC, KH, JK, EO-J, JSe were co-applicants on the original funding application. MR, FVL-W, RC-J, LA, SC, DF, KH, JK, GM, EO-J, RDP, JSa, JSe and TS interpreted the results of the analysis. RC-J and RDP were responsible for producing the tables. MR, FVL-W, RC-J and RDP drafted the study report. MR, FVL-W, RC-J, LA, SC, DF, KH, JK, GM, EO-J, RDP, JSa, JSe and TS reviewed manuscript drafts, revised for important intellectual content and approved the final version.

**Funding** This project was funded by the National Institute for Health Research Public Health Research (NIHR PHR) Programme (reference:11/3002/11).

**Disclaimer** The funders had no role in considering the study design or in the collection, analysis, interpretation of data, writing of the report, or decision to submit the article for publication. The views and opinions expressed therein are those of the authors and do not necessarily reflect those of the NIHR PHR Programme or the Department of Health.

**Competing interests** None declared.

**Patient consent for publication** Not applicable.

**Ethics approval** Ethical approval for this study was granted by the Health Research Authority research ethics committee (Wales REC 3; REC reference 14/WA10062) and section 251 support was provided by the Confidentiality Advisory Group (CAG reference CAG 10-08(b)/2014).

**Provenance and peer review** Not commissioned; externally peer reviewed.

**Data availability statement** Data are available on reasonable request. Anyone wishing to access the linked datasets for research purposes should apply via the CAG to the Health Research Authority to access patient identifiable data without consent and then to the NHS Digital, National Pupil Database and the abortion statistics – Department for Health and Social Care. In the first instance, enquiries about access to the data should be addressed to Professor Mike Robling, Centre for Trials Research, Cardiff University.

**ORCID iDs**
Michael Robling http://orcid.org/0000-0002-1004-036X
Fiona V Lugg-Widger http://orcid.org/0000-0003-0029-9703
Eleri Owen-Jones http://orcid.org/0000-0003-0850-4724
Rhys D Pockett http://orcid.org/0000-0003-4135-7383
Jeremy Segrott http://orcid.org/0000-0001-6215-0870
Thomas Slater http://orcid.org/0000-0003-3840-2454

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
