## [Reviewer comments · BMJ Open]

ARTICLE DETAILS

TITLE (PROVISIONAL)	A nurse-led home-visitation programme for first-time mothers in reducing maltreatment and improving child health and development (BB:2-6): longer-term outcomes from a randomised cohort using data-linkage.
AUTHORS	Robling, Michael; Lugg-Widger, Fiona; Cannings-John, Rebecca; Angel, Lianna; Channon, Sue; Fitzsimmons, Deborah; Hood, Kerensa; Kenkre, Joyce; Moody, Gwenllian; Owen-Jones, Eleri; Pockett, Rhys; Sanders, Julia; Segrott, Jeremy; Slater, Thomas

VERSION 1 – REVIEW

REVIEWER	Olds, DL University of Colorado Denver, Department of Pediatrics I am considered the founder of Nurse-Family Partnership and conducted the the original randomized trials of NFP in the US.
REVIEW RETURNED	22-Mar-2021

GENERAL COMMENTS	This paper presents the results of an age-2-6 follow-up of children and their mothers enrolled in the Building Blocks trial, an RCT of a program of prenatal and infant/toddler home visiting by nurses (Family-Nurse partnership, FNP). The trial focused on young mothers (< 20 years of age) at registration during pregnancy. The investigators used administrative data-linkage to examine children’s involvement in the Social Care system, their rates of injuries and ingestions, school readiness and education outcomes, in addition to maternal rates and timing of subsequent pregnancies and births. In many ways this is an impressive study, technically. It had very high rates of participant matching on the 1647of the mothers and children originally randomized, leading to exceptional preservation of the integrity of the randomization over the 6 years following the child’s birth. Some of the study’s limitation, however, are not adequately discussed and readers need to understand them so they can make informed interpretations of the findings: 1. First, the study compared FNP with “Usual Care.” The investigators did not mask treatment assignment to local health providers, however leading to possible compensatory efforts put forth by heath visitors made possible given that FNP nurses were serving half of the target population, which would free up health visitors to visit the remaining families more frequently. Local midwives and HV in the 18 local health authorities where the study was conducted were informed about which teen mothers were assigned to the intervention versus control conditions, possibly leading them to amplify their efforts with those in the “usual care”
---

	condition. This feature of the design needs to be explicitly stated as a feature of the study design and a possible limitation in the discussion section. 2. It is notable in this regard, that mothers in the usual care group reported receiving an average of 16 visits from Health Visitors. The authors need to report the mean number and range of visits completed for the “Usual Care” group on page 4, lines 27-32. 3. In the authors’ description of the FNP Intervention on page 4 lines 36-44, they present the largest number of possible visits, but not the mean and range of actual number completed. As I understand it, FNP nurses adjust the frequency of visits to families’ needs and availability. Reporting the mean and range of visits completed will enable the reader to interpret the findings in a fully informed way. 4. To assist the reader in grasping the results of this follow-up, I recommend that the authors create two or three tables for the main body of the manuscript that summarize the degree to which the program produced long-term effects on the studies’ primary and secondary outcomes: 1) the child’s designation of being classified as a “Child in Need;” 2) children’s injuries and ingestions; 3) children’s school readiness and educational outcomes; and 4) the timing of subsequent pregnancies and healthcare encounters. The key outcomes get lost in the massive number of outcomes examined and currently reported in the supplement. The reader will appreciate having a clear tabular summary of the outcomes currently reported on pages 11 and 12. The details can be relegated to the supplement. 5. I suggest that the main body of the manuscript show the EYFSP and Key Stage 1 outcomes (including only those findings that result from their adjustment for month of birth). Again, the rest of the analyses can be shown in the supplement. 6. If there are restrictions on the numbers of tables and figures allowed in the main body of the text, I urge the authors to give preference to this set of outcomes and consider moving current table 1, 2, and parts of 3 to the supplement. Again, the reader needs to see the main findings in the main body of the text. 7. The authors refer to the child’s being designated as a CIN, as an objective indicator of child maltreatment on page 3, line 53 and in the tables. While children designated as CIN, on average, have greater needs than those not designated as CIN, these designations are subject to surveillance bias and not, as the authors assert, an objective measure of maltreatment for this reason. The authors acknowledge this in their introduction to the manuscript on page 3 and refer to a paper that discusses surveillance bias in these kinds of studies, but do not provide a citation for the relevant paper.
--	--

REVIEWER	Harron, Katie UCL Great Ormond Street Institute of Child Health Population Policy and Practice I am leading an observational study evaluating the FNP.
REVIEW RETURNED	31-Mar-2021

GENERAL COMMENTS	This paper describes the follow up study to the Building Blocks trial, looking at outcomes in children aged 2-6 years. This is a really important paper and the study has been described very clearly overall. There is a lot of information presented in the appendix, but the main text summarises this well. I just have a few small points for clarification:
--

	Procedure and data sources: Could you be more precise? i.e. identifiers weren't sent to NPD (NPD is a database) – presumably you mean DfE? Which dataset was linked by DHSC? It reads as if data were sent from NHS Digital to ONS, but I don't think this is what you mean. It would be helpful to expand on this (in an appendix if necessary), detailing which datasets were linked to, and where. I'm not clear on what is meant by: We excluded children from analyses of binary outcomes where a child had no event. Do you mean, where there was missing data on the outcome? Month of age is clearly an important covariate, but this isn't included in the description of variables that were adjusted for, in the analysis section. It would be helpful to explain somewhere why some of the developmental outcomes were not tested statistically. There are a couple of formatting errors (“Reference source not found” or [ref], or [insert link]) that need to be altered. The section describing differences according to month of birth is really interesting, but I'm still not clear – if month of birth was balanced between arms, why would you need to adjust for month or birth, and why should adjusting for month or birth result in a change in the effect estimate? The discussion mentions “valid” family nurse visits. How is validity defined? What does mean ES refer to? Reference missing page 16 line 40 (presumably for BB:0-2) Figure 1. The boxes are currently overlapping which makes it difficult to read. Can you give reasons for the 5 ineligible mothers? “Removed as deceased n=1 M+C” – should this be mother deceased? Can you explain somewhere why so few of the mothers were matched with NPD – i.e. this isn't a problem with the matching, but because you didn't attempt to link mothers with their education data, and you would only expect them to link to social care data if they had had an interaction with social care services. How did you use data on social care for mothers? This doesn't seem to be included in Table S1.1 (or should this come under Health and Social Care resource use?) Not clear why the sample size calculation is repeated in the appendix. It is interesting to see that so many mothers electively withdrew from BB0-2 but consented for BB2-6? Do you know the reasons why they withdrew initially? The percentages given in Table S6.1 are not of the total in each arm, but within each category, i.e. 63.0% of those who had a CIN referral were referred due to abuse/neglect, NOT 63.0% of all children in the UC arm. I think this would also be helpful to give the
--	--

	percentages of the total in each arm (or at the very least, give the denominator in the outcome heading row so that this is clear). Table S6.2. What is “rate of absence rate”? Table S6.3 refers to a proportion but percentages are presented. This table should include the denominators. Table S7.5 – Could you make the difference between the two columns of adjusted ORs clearer in the headings – it took me a while to figure out what the difference was.
--	---

VERSION 1 – AUTHOR RESPONSE

Reviewers comments	Response
Reviewer: 1 - Dr. DL Olds, University of Colorado Denver	
This paper presents the results of an age-2-6 follow-up of children and their mothers enrolled in the Building Blocks trial, an RCT of a program of prenatal and infant/toddler home visiting by nurses (Family-Nurse partnership, FNP). The trial focused on young mothers (< 20 years of age) at registration during pregnancy. The investigators used administrative data-linkage to examine children’s involvement in the Social Care system, their rates of injuries and ingestions, school readiness and education outcomes, in addition to maternal rates and timing of subsequent pregnancies and births.	No response
In many ways this is an impressive study, technically. It had very high rates of participant matching on the 1647of the mothers and children originally randomized, leading to exceptional preservation of the integrity of the randomization over the 6 years following the child’s birth.	No response
Some of the study’s limitation, however, are not adequately discussed and readers need to understand them so they can make informed interpretations of the findings:	
1. First, the study compared FNP with “Usual Care.” The investigators did not mask treatment assignment to local health providers, however leading to possible compensatory efforts put forth by heath visitors made possible given that FNP nurses were serving half of the target population, which would free up health visitors to visit the remaining families more frequently. Local midwives and HV in the 18 local health authorities where the study was conducted were informed about which teen mothers were assigned to the intervention versus control conditions, possibly leading them to amplify their efforts with those in the “usual care” condition. This feature of the design needs to be explicitly stated as a feature of the study design and a possible limitation in the discussion section.	We have added a narrative on this including citing our previous work on this point to the Discussion. This is in addition to providing metrics in the Methods section.
2. It is notable in this regard, that mothers in the usual care group reported receiving an average of 16 visits from Health Visitors. The authors need to report the mean number and range of visits completed for the “Usual Care” group on page 4, lines 27-32.	We have added detail of the maternally reported contacts with both health visitors and midwives to the Methods section.
3. In the authors’ description of the FNP Intervention on page 4 lines 36-44, they present the largest number of possible visits, but not the mean and range of actual number completed. As I understand it, FNP nurses adjust the frequency of visits to families’ needs and availability. Reporting the mean and range of visits completed will enable the reader to interpret the findings in a fully informed way.	We have added detail of the number of FNP programme reported valid visits by delivery phase to the Methods section description.
4. To assist the reader in grasping the results of this follow-up, I recommend that the authors create two or three tables for the main body of	We agree that some re-presentation of the study outcomes is desirable.

the manuscript that summarize the degree to which the program produced long-term effects on the studies' primary and secondary outcomes: 1) the child's designation of being classified as a "Child in Need;" 2) children's injuries and ingestions; 3) children's school readiness and educational outcomes; and 4) the timing of subsequent pregnancies and healthcare encounters. The key outcomes get lost in the massive number of outcomes examined and currently reported in the supplement. The reader will appreciate having a clear tabular summary of the outcomes currently reported on pages 11 and 12. The details can be relegated to the supplement.	In addition to the original Tables 3 and 4, supplementary tables S6.1 and S6.2 represent the totality of statistically assessed a priori primary and secondary outcomes. We have integrated these into the main body of the manuscript (ie as three tables). We have further placed all descriptive only, subgroup and all exploratory analyses in the appendices. We consider this reflects better the study's analysis plan and makes the main manuscript easier to navigate whilst still including all results. Our original presentation of tabulated outcomes was driven in part by formatting requirements of the journal regarding table sizes (hence much material being placed in appendices). Although even longer tables are currently being published in the journal as part of the main paper, we are aware that size of tables may be a consideration for the journal. If so, we would remove problematic tables to the appendix if required.
5. I suggest that the main body of the manuscript show the EYFSP and Key Stage 1 outcomes (including only those findings that result from their adjustment for month of birth). Again, the rest of the analyses can be shown in the supplement.	The analysis of EYFSP and KS1 data adjusting for month of birth was exploratory which is why the data are tabulated and presented in the supplementary materials. However, these results are presented in the narrative of the Results section and which we consider represents both the research plan and also the relevance of the adjusted data.
6. If there are restrictions on the numbers of tables and figures allowed in the main body of the text, I urge the authors to give preference to this set of outcomes and consider moving current table 1, 2, and parts of 3 to the supplement. Again, the reader needs to see the main findings in the main body of the text.	We have moved more of the findings previously located in the supplementary materials to the main body of the manuscript (see above). We have moved Table 1 to the appendix. We think it important to retain Table 2 (baseline characteristics of families by trial arm) in the main section and now this becomes Table 1. Table 3 has been integrated into new tables describing study outcomes (as noted above) and we consider these should remain in the main body of the manuscript.
7. The authors refer to the child's being designated as a CIN, as an objective indicator of child maltreatment on page 3, line 53 and in the tables. While children designated as CIN, on average, have greater needs than those not designated as CIN, these designations are subject to surveillance bias and not, as the authors assert, an objective measure of maltreatment for this reason. The authors acknowledge this in their introduction to the manuscript on page 3 and refer to a paper that discusses surveillance bias in these kinds of studies, but do not provide a citation for the relevant paper.	The identification and subsequent determination of maltreatment will involve several steps which can be influenced by subjective and variable processes even if these culminate in harder measurements (including medical imagery, for example). We think that verified is a useful term to use to describe outcomes such as designation as being in Need as they have been generated by a statutory process. We have added a commentary on this in Discussion and included references as appropriate.

Reviewer: 2 Dr. Katie Harron, UCL Great Ormond Street Institute of Child Health Population Policy and Practice	
This paper describes the follow up study to the Building Blocks trial, looking at outcomes in children aged 2-6 years. This is a really important paper and the study has been described very clearly overall. There is a lot of information presented in the appendix, but the main text summarises this well. I just have a few small points for clarification:	We thank the reviewer for her comments
Procedure and data sources: Could you be more precise? i.e. identifiers weren't sent to NPD (NPD is a database) – presumably you mean DfE? Which dataset was linked by DHSC? It reads as if data were sent from NHS Digital to ONS, but I don't think this is what you mean. It would be helpful to expand on this (in an appendix if necessary), detailing which datasets were linked to, and where.	We agree and have corrected these references to NPD accordingly. We have clarified this in the main document text: NHS Digital also provided a linked Hospital Episode Statistics (HES) -Office for National Statistics (ONS) extract for mortality data.
I'm not clear on what is meant by: We excluded children from analyses of binary outcomes where a child had no event. Do you mean, where there was missing data on the outcome?	Yes if a child had incomplete follow-up period and had not experienced an outcome, then they would be excluded. I have clarified this sentence.
Month of age is clearly an important covariate, but this isn't included in the description of variables that were adjusted for, in the analysis section.	This has been added in "An additional post hoc sensitivity analysis adjusted for the child's month of birth (categorised into quarter of birth) in the EYFSP and KS1 outcomes."
It would be helpful to explain somewhere why some of the developmental outcomes were not tested statistically.	We have added some text to the Methods section describing our approach (ie to avoid a large number of comparative analyses, to address outcomes that may reflect a very small number of events or high rates of missingness). We have described in table footnotes (in the outcomes tables in supplementary section where these results are now presented) outcomes denoted as "NT" as "NT = Not tested, descriptive only." Additionally, for the KS1 writing assessment, the statistical analysis section states that: "However, following changes made within the 2017/18 writing assessment (such that judgements in 2018 are not directly comparable to those made using the previous interim frameworks in 2016/2017), writing was only statistically assessed using 2016/17 data and the data from 2017/18 were presented descriptively."
There are a couple of formatting errors ("Reference source not found" or [ref], or [insert link]) that need to be altered.	We have edited to include now the insert link reference.
The section describing differences according to month of birth is really interesting, but I'm still not clear – if month of birth was balanced between arms, why would you need to adjust for month or birth, and why should adjusting for month or birth result in a change in the effect estimate?	What is observed from the writing assessment is the increase in the rates of children reaching at least the expected standard between the two academic years (from 45.7% in 2016/17 to 66.5% in

	2017/18). This is mainly explained by the distribution of births for the children for each academic year. Children with KS1 results available in the academic year 2016/17 were born between October 2009 and August 2010, but the distribution of month of birth was skewed towards the summer months, whereas children appearing on the 2017/18 academic year were born between September 2010 and February 2011. There is a nationally shown relationship between rates of children reaching the expected standard and month of birth, for example a 17% difference in reading level for children born in August and September. So whilst the treatment effect for the 2016/17 cohort data is still valid, what it does highlight is that month of birth has to be taken into account as important moderator of educational outcome. The original manuscript provided some of the detail above and we have added a small section to the Discussion to better clarify this point.
The discussion mentions “valid” family nurse visits. How is validity defined?	Valid visits were defined as follows:  1. UK001 visit form is completed, 2. the visit is completed, 3. Visit is at least 15 minutes duration, 4. Client is present (unless father is present and is listed as main carer) 5. first scheduled visit of the day Full details of how validity of visits was established are provided in the original trial report (which we have now cited). Validity criteria were those established by the FNP programme.
What does mean ES refer to?	This has been changed in the discussion to effect size.
Reference missing page 16 line 40 (presumably for BB:0-2)	Yes - we have corrected this.
(i) Figure 1. The boxes are currently overlapping which makes it difficult to read. (ii) Can you give reasons for the 5 ineligible mothers? (iii) “Removed as deceased n=1 M+C” – should this be mother deceased? (iv) Can you explain somewhere why so few of the mothers were matched with NPD – i.e. this isn’t a problem with the matching, but because you didn’t attempt to link mothers with their education data, and you would only expect them to link to social care data if they had had an interaction with social care services.	(i) We have updated the flow chart and provided in a pdf. (ii) The 5 mothers were identified to be ineligible as they did not meet the trial eligibility criteria: one woman was deemed not to be Gillick competent, one woman was identified as not pregnant at the first scan, 3 women were registered with a GP outside of the study area. We have added this clarification to the Results section and cited the original trial paper. (iii) This is one mother child dyad / family was subsequently removed. (iv) Mothers had a low rate of matching owing to having discontinued their education,

(v) How did you use data on social care for mothers? This doesn't seem to be included in Table S1.1 (or should this come under Health and Social Care resource use?)	very few mothers, compared with children, were matched to any NPD data set (19% vs. 97%, respectively); these events were based mainly on the CIN data set. This sentence has been added. (v) The maternal data on social care by the time of recruitment was used only in an exploratory subgroup analyses examining maternal care status and duration of maternal care and domestic abuse self-reported. The detail of this is included in section 7 in the supplementary material under: Care-experienced mothers and CIN status.
Not clear why the sample size calculation is repeated in the appendix.	It has been removed from the supplementary material.
It is interesting to see that so many mothers electively withdrew from BB0-2 but consented for BB2-6? Do you know the reasons why they withdrew initially?	Reasons for electively withdrawing from the trial is detailed in the main trial paper (which we have now cited in the Results section as appropriate). These reasons included no longer wishing to take part, not wanting to commit to the FNP Programme, not wanting to take part in the trial as not allocated to FNP, moving out of area, and considering placing the baby for adoption.
The percentages given in Table S6.1 are not of the total in each arm, but within each category, i.e. 63.0% of those who had a CIN referral were referred due to abuse/neglect, NOT 63.0% of all children in the UC arm. I think this would also be helpful to give the percentages of the total in each arm (or at the very least, give the denominator in the outcome heading row so that this is clear).	Denominators of children in need added to the table.
Table S6.2. What is "rate of absence rate"?	This has been changed to Absence rate and Median (25th to 75th centiles) added.
Table S6.3 refers to a proportion but percentages are presented. This table should include the denominators.	As suggested the numbers have been added to this table for transparency.
Table S7.5 – Could you make the difference between the two columns of adjusted ORs clearer in the headings – it took me a while to figure out what the difference was.	The footnote has been made clearer for all tables that additionally adjust for month of birth.

VERSION 2 – REVIEW

REVIEWER	Olds, DL University of Colorado Denver, Department of Pediatrics I am the founder of Nurse-Family Partnership and led the original trials of the program in the US. The University of Colorado and I own the intellectual property on which FNP is based and charge modest licensing fees to countries and organizations outside of the US for the use of these materials. The University of Colorado also charges those countries and organizations consultation fees for guidance in delivering the program. The aim is to ensure quality program implementation and protection of the intellectual property. Those fees help cover my university salary, but do not augment my salary.
---

GENERAL COMMENTS

The current draft is a significant improvement over first version, but I have a number of remaining concerns:

1. The report needs to acknowledge, in the interest of transparency, that treatment assignment was not masked to usual-care providers.
2. Throughout this report the authors use the word “objective” to describe the CIN-related outcomes as a reflection of child maltreatment. While it’s true that these outcomes are not based upon parent report, these features of child protection systems are far from “objective” and vary dramatically cross-nationally and within countries.¹⁻³ Features of these systems shape providers’ ways of managing their concerns about children’s needs. What is examined in this report is the degree to which NFP-nurses and usual-care providers observed children in need and made referrals to CIN/child protection services to address those perceived needs. The report needs to be framed in a way that reflects this insight. Given FNP nurses’ deep involvement with women during pregnancy and the early months of the child’s life, they are in a position to observe family needs and child wellbeing quite thoroughly.
3. Note that the number of referrals to CIN during pregnancy and the first year of the child’s life are less than 5 for treatment x age subclasses. The authors can reduce this problem by combining prenatal and 0-1 ages.
4. Given the thoroughness with which the CIN data were reported, I was surprised to see that there were 277 referrals of children for CIN evaluations that were not deemed sufficiently serious to warrant confirmation and that these referrals were not reported by treatment condition. The full interpretation of this study depends on knowing how many of such referrals were made for CIN by treatment condition. These cases may reflect FNP nurses’ concerns about vulnerable families and their commitment to ensuring their protection. These data should be broken down by the ages at which referrals were made and the source of referrals (i.e., individual, health services, etc.), just as the referrals of confirmed cases were classified. This, of course, is one aspect of the surveillance-bias phenomenon the authors refer to in the introduction of this report. FNP nurses are guided to work with safeguarding nurses and other providers to ensure child protection. Reporting these rates by treatment will help address this issue. Note that the study protocol says that investigators will compare severity of referral, but this is not done. Looking at referrals not acted upon will give insight to this issue.
5. Use of the adjective “objective” conveys an inaccurate characterization of maltreatment reports, as FNP nurses have deeper involvement with the families they visit and are likely to identify maltreatment at lower thresholds of severity. Again, this is why reporting on the CIN referrals that were not acted upon is so crucial.
6. The shorter length of time that FNP-visited children were in CIN is likely a reflection of FNP families being referred for care at lower thresholds of severity.
7. Much of this report addresses how children and families were processed within the system to ensure their safety and promote their wellbeing.
8. The closest thing to an “objective” measure of maltreatment is death due to preventable causes, especially before children are mobile; short of death, injury or ingestion before children are

mobile that is sufficiently severe to require hospitalization is an objective indicator of the quality of parental care. Severity of injury/ingestion is further marked by the number of days very young children were hospitalized. Many very young (pre-mobile) children hospitalized for injury are at risk of dying without timely care. Hospitalizations for injury once children are mobile are shaped by other contextual factors to a greater degree and influenced less by parental care.

9. The authors have listed hospitalization for injury as an outcome in their trial protocol and the data are available through HES on an annual basis, including admission and discharge dates. The authors have this outcome in their HES data but have not reported the days children were hospitalized with injuries by year of life in this report. Readers need to know the degree to which FNP affected this outcome, given its objectivity in the BB 0-6 study and the presence of treatment-control differences on this outcome in an earlier US trial. The authors should report this outcome by year of the child's life and length of hospitalizations. Including this outcome will strengthen the objectivity of measurement - both of child health functioning and cost given that lengths of stay also provide a more accurate estimate of cost.

10. Note that A&E attendances and primary care encounters for injuries reflect both the incidence and severity of injuries, but also FNP nurses' encouraging families to have relatively minor injuries evaluated to rule out more serious internal damage. Their greater involvement of FNP nurses with families in these early months of life puts them in a position to guide families on these issues. Such encounters need to be interpreted with this more nuanced understanding of how FNP nurses guide parents in addressing injuries.

11. The current report shows A&E and hospitalizations as a categorical variable (yes or no). The fundamental differences in numbers and types of encounters need to be elaborated. Note that this broad treatment of healthcare encounters stands in sharp contrast to the reports' treatment of CIN-related services, which are classified in exquisite detail.

12. While the educational effects are relatively small, they are consistent across measures and of clear social and economic importance. They hold promise of future benefits. Note that some effects only reach the level of trends (such as children being classified as having Special Educational Needs (SEN)) but are important, and again are consistent with other objective measures of child functioning. They reflect cost and developmental functioning, and indicate the range of directly measured benefits to children. The effects only present as trends deserve to be acknowledged given their consistency.

13. Given that the KS1 outcomes are known to be affected by month of birth, the analysis of KS1 outcomes that controls for month of birth should not be treated as the primary analysis, not exploratory. It provides the best estimate of FNP effects and should be presented as analysis of this set of outcomes.

14. The qualification of results in the discussion section referencing the adjustment for month of birth gives the impression that FNP effects differed by month of birth, when the more correct statement is that program effects were observed once a more comprehensive adjustment was made for a variable that biased the treatment contrast.

16. Please note that references are not correct.

17. Note that maternal educational achievement appears to be missing because the mothers were no longer in education.

	Examining the full range of educational outcomes for even those mothers who are no longer in education is warranted and important for this trial. Can incomplete records be examined? 18. Please place the order of UC and FNP services consistently in description of services/tx groups (pages 5-6). 19. Note that length of visits was essentially identical to those found in US trials. References 1. Merkel-Holguin, Lisa, Fluke, John D., Krugman, Richard (Eds). National Systems of Child Protection: Understanding the International Variability and Context for Developing Policy and Practice. Springer International Publishing AG (2019). 2. Tung, GJ, Williams, VN, Ayele, R, Shimasaki, S, Olds, D. "Characteristics of Effective Collaboration: A study of Nurse-Family Partnership and Child Welfare." Child Abuse & Neglect. 2019;95: 104028. https://doi.org/10.1016/j.chiabu.2019.104028 3. Williams, VN, Ayele, R, Shimasaki, S, Tung, GJ, Olds, D. "Risk assessment practices among home visiting nurses and child protection caseworkers in Colorado, United States: A qualitative investigation." Health & Social Care in the Community. 2019;27(5): 1344-52. doi.org/10.1111/hsc.12773
--	--

REVIEWER	Harron, Katie UCL Great Ormond Street Institute of Child Health Population Policy and Practice I am lead for a separate observational study evaluating the FNP in England, and some of the authors of this paper are collaborators on that study.
REVIEW RETURNED	06-Jul-2021

GENERAL COMMENTS	I am satisfied with the responses to my previous comments.
--

VERSION 2 – AUTHOR RESPONSE

#	Comment	Response
1	The report needs to acknowledge, in the interest of transparency, that treatment assignment was not masked to usual-care providers.	The manuscript currently states that the trial was open-label and includes a paragraph in Discussion describing the potential for bias in such circumstances. We consider that this is fully transparent and includes citations to the full original trial protocol and report.
2	Throughout this report the authors use the word "objective" to describe the CIN-related outcomes as a reflection of child maltreatment. While it's true that these outcomes are not based upon parent report, these features of child protection systems are	We have previously clarified the sense in which we use the term objective to describe those outcomes which are not simply respondent self-report. Although we believe this description still holds true, we have

	far from “objective” and vary dramatically cross-nationally and within countries.1-3 Features of these systems shape providers’ ways of managing their concerns about children’s needs. What is examined in this report is the degree to which NFP-nurses and usual-care providers observed children in need and made referrals to CIN/child protection services to address those perceived needs. The report needs to be framed in a way that reflects this insight. Given FNP nurses’ deep involvement with women during pregnancy and the early months of the child’s life, they are in a position to observe family needs and child wellbeing quite thoroughly	amended the descriptors we have used to avoid any confusion. We agree that the ascertainment and recording of outcomes captured in administrative systems will reflect differences in inter-/intra-professional (and systemic) knowledge, skills, judgement and policies. We have added a section to the Discussion to address this point.
3	Note that the number of referrals to CIN during pregnancy and the first year of the child’s life are less than 5 for treatment x age subclasses. The authors can reduce this problem by combining prenatal and 0-1 ages.	Combining these two categories in re-presenting these data (in Table S6) will add no further information as the proposed aggregate values are already obtainable from the column counts (ie there is a column total for each treatment group and there are two categories – prenatal and <1 yrs – with small numbers). In addition, it would be unusual to combine pre-birth and 0-1 ages. There is a tendency towards having pre-birth as a distinct category as it is not possible to secure a court order until someone has been born. It is also is not possible to combine 0-1 and 1-2 as it then identifies the small numbers in the pre-birth category.
4	Given the thoroughness with which the CIN data were reported, I was surprised to see that there were 277 referrals of children for CIN evaluations that were not deemed sufficiently serious to warrant confirmation and that these referrals were not reported by treatment condition. The full interpretation of this study depends on knowing how many of such referrals were made for CIN by treatment condition. These cases may reflect FNP nurses’ concerns about vulnerable families and their commitment to ensuring their protection. These data should be broken down by the ages at which referrals were made and the source of referrals (i.e., individual, health services, etc.), just as the referrals of confirmed cases were classified. This, of course, is one aspect of the surveillance-bias phenomenon the authors refer to in the introduction of this report. FNP nurses are guided to work with safeguarding nurses and other providers to ensure child protection. Reporting these rates by treatment will help address this issue. Note	The figure of 277 referrals not resulting in children being assessed as in need of additional support (ie a Child in Need) includes repeat referrals for some children. There were 693 unique referrals involving 414 children (FNP n=206, UC n=208) and of these, 323 children were assessed as in need (FNP n=161, UC n=162). The number of children referred to CSC and assessed as not requiring additional support were FNP (n=45) and UC (n=46). We have edited the manuscript to make this much clearer, as we would agree that this was not either originally included or sufficiently clear. This does not suggest that children are being referred to CSC at lower thresholds (and resulting in no further action once assessed)

	that the study protocol says that investigators will compare severity of referral, but this is not done. Looking at referrals not acted upon will give insight to this issue.	or the presence of surveillance bias. We have commented on this briefly in Discussion.
5	Use of the adjective “objective” conveys an inaccurate characterization of maltreatment reports, as FNP nurses have deeper involvement with the families they visit and are likely to identify maltreatment at lower thresholds of severity. Again, this is why reporting on the CIN referrals that were not acted upon is so crucial.	We have responded to this point above (in response to query 2), in the text added to the Discussion section and made it clear what we mean by the term objective. Referrals to CSC not taken further forward is picked up on addressed in our response to query 4.
6	The shorter length of time that FNP-visited children were in CIN is likely a reflection of FNP families being referred for care at lower thresholds of severity	We agree that a difference in length of time being registered as a CIN could reflect a lower threshold for referral. Additionally, it could reflect greater capacity of the family to respond effectively to the additional support offered or greater confidence within social services in referring children back with no further enhanced support. All of these explanations are potential positive outcomes of FNP intervention. However, the difference between treatment arms in duration of being CIN is not statistically different and therefore, we have not commented further. Nevertheless, the likelihood of referral at lower thresholds is addressed in our responses to other queries (eg query 2 in text added to Discussion).
7	Much of this report addresses how children and families were processed within the system to ensure their safety and promote their wellbeing.	No comment suggested.
8	The closest thing to an “objective” measure of maltreatment is death due to preventable causes, especially before children are mobile; short of death, injury or ingestion before children are mobile that is sufficiently severe to require hospitalization is an objective indicator of the quality of parental care. Severity of injury/ingestion is further marked by the number of days very young children were hospitalized. Many very young (pre-mobile) children hospitalized for injury are at risk of dying without timely care. Hospitalizations for injury once children are mobile are shaped by other contextual factors to a greater degree and influenced less by parental care	We have run further exploratory descriptive analyses (including confidence intervals) which describe the breakdown of hospital inpatient stays consequent to an injury or ingestion. These are broken down by categories showing the stays occurring in the first year of life, the second year of life and then the subsequent period after the end of the programme to age 6. We have reported the duration of hospital / inpatient stay by study arm for each of these sub-groups. These are described in the main results section with the tables in the supplementary appendix. We have added a paragraph to the Discussion which reflects on results for length of hospital stay reported in the US and in our study, and also what factors may be associated with length of hospital stays as an outcome (for example, LOS as a marker of maltreatment, of clinical severity and other factors).

9	The authors have listed hospitalization for injury as an outcome in their trial protocol and the data are available through HES on an annual basis, including admission and discharge dates. The authors have this outcome in their HES data but have not reported the days children were hospitalized with injuries by year of life in this report. Readers need to know the degree to which FNP affected this outcome, given its objectivity in the BB 0-6 study and the presence of treatment-control differences on this outcome in an earlier US trial. The authors should report this outcome by year of the child's life and length of hospitalizations. Including this outcome will strengthen the objectivity of measurement - both of child health functioning and cost given that lengths of stay also provide a more accurate estimate of cost.	Please note our response above which addresses the analysis of length of hospital stay. In our added section to the Discussion we have also discussed the nature of the measure of length of hospital stay. Factors other than severity of clinical presentation will affect length of stay although the nature of the clinical presentation will likely be a substantive part of that. Whilst the relative subjectivity or objectivity of different study outcomes are important, key is how this then impacts on validity for measuring intended constructs.
10	Note that A&E attendances and primary care encounters for injuries reflect both the incidence and severity of injuries, but also FNP nurses' encouraging families to have relatively minor injuries evaluated to rule out more serious internal damage. Their greater involvement of FNP nurses with families in these early months of life puts them in a position to guide families on these issues. Such encounters need to be interpreted with this more nuanced understanding of how FNP nurses guide parents in addressing injuries.	We agree with this assessment of the nurse role and likely impact and have commented on this in previous work. However, as this manuscript includes no data on primary care encounters and no difference between trial arms for ER attendance we have not commented further on this point.
11	The current report shows A&E and hospitalizations as a categorical variable (yes or no). The fundamental differences in numbers and types of encounters need to be elaborated. Note that this broad treatment of healthcare encounters stands in sharp contrast to the reports' treatment of CIN-related services, which are classified in exquisite detail.	We have added (our previously published) tables showing ER attendances and hospital admissions by category of presentation to the supplementary appendix.
12	While the educational effects are relatively small, they are consistent across measures and of clear social and economic importance. They hold promise of future benefits. Note that some effects only reach the level of trends (such as children being classified as having Special Educational Needs (SEN)) but are important, and again are consistent with other objective measures of child functioning. They reflect cost and developmental functioning, and indicate the range of directly measured benefits to children. The effects only present as trends deserve to be acknowledged given their	We do not support the use of 'trends' as the most appropriate basis for presenting results from this trial cohort especially given the very large number of comparative analyses included in the main results. We also consider this unnecessary given that all planned analyses (plus additional exploratory) are being reported allowing full transparency to the reader.

	consistency.	
13	Given that the KS1 outcomes are known to be affected by month of birth, the analysis of KS1 outcomes that controls for month of birth should not be treated as the primary analysis, not exploratory. It provides the best estimate of FNP effects and should be presented as analysis of this set of outcomes	The findings are presented in accordance with our a priori analysis plan. The additional analysis and the rationale for this further adjustment is described in full. This provides complete transparency about the study.
14	The qualification of results in the discussion section referencing the adjustment for month of birth gives the impression that FNP effects differed by month of birth, when the more correct statement is that program effects were observed once a more comprehensive adjustment was made for a variable that biased the treatment contrast	We agree, was not the impression intended and have removed the sentence.
16	Please note that references are not correct.	We have checked and updated / edited as appropriate.
17	Note that maternal educational achievement appears to be missing because the mothers were no longer in education. Examining the full range of educational outcomes for even those mothers who are no longer in education is warranted and important for this trial. Can incomplete records be examined?	The study was funded to assess programme impact upon child maltreatment outcomes. Maternal educational achievement was never an outcome of the follow-up study and as such maternal education outcomes were not requested from the data provider (NPD).
18	Please place the order of UC and FNP services consistently in description of services/tx groups (pages 5-6).	We have edited this to ensure consistency.
19	Note that length of visits was essentially identical to those found in US trials.	We have described how duration of observed valid visits compared to the programme specified duration and therefore one quality indicator of programme delivery in England. We did not find visit duration reported in the primary publication for the Elmira and Memphis trials (Olds et al Pediatrics 1986 77; 16-28 and Olds et al Pediatrics 1986 78; 65-78; Kitzman et al 1997) but can include a reference to these data if we have missed this or this information is available elsewhere.

VERSION 3 – REVIEW

REVIEWER	Olds, DL University of Colorado Denver, Department of Pediatrics
-----------------	---

	I conducted a series of randomized controlled trials of Nurse-Family Partnership, as the program is named in the US. The center I direct receives licensing fees for implementation of the program and contracts to support the program in international contexts. I do not receive compensation for the program beyond my university salary, although I periodically receive honoraria for speaking about the program and the prevention of maternal, child, and family health and developmental problems.
REVIEW RETURNED	17-Aug-2021

GENERAL COMMENTS	The current draft is a significant improvement over the previous version, but there are additional issues that need to be resolved: 1. While it's true that the authors have consistently referred to this work as an "Open Label" trial, the usual meaning of Open Label is that participants and providers are aware of some specialized treatment like a drug or medical device that does not directly compete with the usual-care providers themselves in estimating their effectiveness. In the context of this study, the meaning of "Open" needs to be revealed more thoroughly, given that FNP nurses assumed the role of the Health Visitor in the study sites.¹ This reduced scope of Health Visitor practice in the FNP sites, put them in a position to visit those in "Usual Care" more frequently and with more focus than typically is possible for Health Visiting in the UK. While the authors have examined this issue in a paper they have published on this topic,² it is not clear whether usual care from Health Visitors in this study was representative of the way Health Visitors delivered the program to this population at the start of the trial. This feature of the design raises questions about its external validity, given that Usual Care was potentially augmented by having FNP nurses take over the role of Health Visitors for half of the targeted population. While the current version of this manuscript goes further in acknowledging this limitation, I have related concerns about the reports of how many visits were completed by Health Visitors (herein referred to as "Specialist Public Health Nurses") to Usual Care and FNP-visited families. 2. In reporting characteristics of Usual Care in the current manuscript, it will be important to note that estimates of Midwife and Health-Visitor encounters were based upon participant reports (and subject to challenges with memory and uncertainty about how respondents interpreted questions regarding these encounters). Participants' reports of encounters were based on interviews conducted at ages 6, 12, and 18 months postpartum but the FNP program goes through age 24 months. Note that other aspects of the interview conducted to measure Usual Care were conducted at 6, 12, 18, and 24 months postpartum.² Can the authors carry out estimates of Specialist Public Health Nurses (page 5, lines 57-59, and page 6, lines 4 and 5) by including data gathered at 24 months? (FNP encounters, as reported in the current paper, were based upon data from the FNP information system and go through 24 months.) These differences in sources of data need to be acknowledged in the current report given that FNP-visited and Usual-Care mothers reported nearly identical numbers of home visits. This seems odd given that FNP nurses were charged with assuming the Health Visitor role in the study sites.¹ What did the Specialized Public Health Nurses address in their relatively frequent home visits to families assigned to receive Health Visiting by FNP nurses? I tried to access the online supplement that laid out the questions examined in the "Usual-Care" paper² to address these questions, but the link is not working.
--

	3. Given that FNP nurses assumed the Health-Visitor role, what did the Specialist nurses do with FNP-visited families in their homes? Weren't they duplicating the work of the FNP nurses? In what way did they provide added service to the FNP-visited families? Did they spend the same amount of time when visiting Usual-Care and FNP-visited families? 4. To some degree, the authors have addressed this inter-related set of concerns in the revised discussion section (page 23 lines 43-60 and page 24, lines 3-11). This revised discussion helps put this study in context, but it does not fully address the issues enumerated in 1 and 2 above. 5. The current manuscript is an improvement over the previous version by giving some insight into the numbers and LOS of hospitalizations for injuries and ingestions by age (0-1 and >1-2 years) and by the thoughtful discussion of this issue in the discussion section. The data indicate that some admissions and discharges took place on the same day, however, suggesting that such admissions were not as serious as those that involved longer stays. A strong case can be made for excluding these less-serious cases (marked by same-day discharge.) The hypothesis, formulated on the basis of previous NFP trials, is that very young FNP-visited children were less likely to have been harmed due to problematic parental behavior. The analyses conducted so far appear to be by quantile regression, which truncates the range of values (and has included same-day discharges). A robust case can be made for analyzing days hospitalized in years 0-1 and years >1-2 (as well as combined as they were in the Memphis NFP trial) using Poisson regression given that extreme values have clinical meaning. In addition, the reader should be able to examine the specific ICD-10 codes associated with hospitalizations in the 0-2 age range (as the authors have done for all hospitalizations over the entire period in Table S8). Such analyses hold the potential for revealing in a much more objective way whether the program prevented serious harm to children at a highly vulnerable age -- when such harm is likely due to abuse or neglect and when it is likely to have life-long impacts. Given the enormous investments made in this study, the investigators have a responsibility to share these findings with the scientific and public- policy communities in determining the degree to which such effects are replicated. 6. The authors' response to my suggestion that they combine prenatal and 0-1 referrals to CIN in Table S6 was that there is no reason to combine pregnancy and 0-1 age categories because the data can be easily counted. Perhaps so, but the reader needs to have such data presented in a more accessible way. The 13 FNP cases in these combined age ranges (8.1%) compared to 5 UC (3.1%) is an important difference that will help with the interpretation of findings. Prenatal referrals probably reflect providers' concerns about profound mental illness, substance abuse, domestic violence, etc., conditions that are relatively unalterable and put the child in imminent harm. Such children need protection. Knowing the rate of referrals for CIN in these very earliest months is critical to revealing whether and how having FNP at this phase in life contributes to child protection/safety. 7. In testing for surveillance bias in this trial, analyses could be conducted that would correspond to those conducted in the Elmira trial (reference 20), using the educational data from the BB2-6 study. Given differences in the health systems in Elmira (four decades ago) and the system in England today, use of health records for parallel analyses should be conducted with great circumspection. It's not clear that such data have the same
--	--

	meaning in the English contemporary context. Conducting these types of analyses using the educational data would be a service to those who fund this study and to the broader communities committed to improving maternal and child health. 8. Note on page 3, line 33, the impact of the Elmira program on maltreatment verified reports for the whole sample measured through child-age 15, current reference 30. 9. Note also on page 3, line 33 the impact of NFP in Memphis on days hospitalized for injuries and ingestions through child-age 2, an objective possible indication of maltreatment. (Reference 2). 10. My question about maternal educational achievement is this: Can the rates of missing data/records on maternal educational achievement be used to estimate the degree to which mothers completed educational milestones? Missing data may indicate that mothers were no longer in education when data were gathered, but the available data may reflect how much they achieved while they were enrolled. Those with no current records may have dropped out of education. The authors have gathered these data even though some records are incomplete. Incomplete records may have educational meaning. Given the possible use of these data in estimating the effect of the program on this aspect of maternal functioning, it may be worth exploring along the lines I have suggested, or at least explaining whether this is possible. 11. Note that length of visits was essentially identical to those found in US trials, that is 75-90 minutes. 3 References 1. https://bmcpediatr.biomedcentral.com/articles/10.1186/1471-2431-13-114#citeas 2. https://bmjopen.bmj.com/content/8/5/e020152 3. https://pubmed.ncbi.nlm.nih.gov/12387552/
--	---

VERSION 3 – AUTHOR RESPONSE

Response to reviewer comments (BB2 main manuscript to BMJOpen)

Nov 2021

We thank the reviewers for all their time, expertise and comments on our manuscript and for facilitating its revision.

#	Review comment	Author response
1	While it's true that the authors have consistently referred to this work as an "Open Label" trial, the usual meaning of Open Label is that participants and providers are aware of some specialized treatment like a drug or medical device that does not directly compete with the usual-care providers themselves in estimating their effectiveness. In the context of this study, the meaning of "Open" needs to be revealed more thoroughly, given that FNP nurses assumed the role of the Health Visitor in the study sites.¹ This reduced scope of Health Visitor practice in the FNP sites, put them in a position to visit those in "Usual Care" more frequently and with more focus than typically is possible for Health Visiting in the UK. While the authors have examined this issue in a paper they have published on this topic,² it is not clear	As the reviewer notes we provided added focus on this point in our revised manuscript including the potential for service professionals to enhance provision for woman allocated to the usual care arm (ie the implications of an open label approach). The minimal reduction in client caseload for individual HVs in trial sites associated with some women entering FNP is unlikely to have afforded much opportunity to importantly enhance their offer. The comparative caseload figures we now provide talks to that but of course such data are imperfect in providing a definitive comparison. However, our qualitative work at the time of the trial (eg focus group data from HVs at trial sites)

	whether usual care from Health Visitors in this study was representative of the way Health Visitors delivered the program to this population at the start of the trial. This feature of the design raises questions about its external validity, given that Usual Care was potentially augmented by having FNP nurses take over the role of Health Visitors for half of the targeted population. While the current version of this manuscript goes further in acknowledging this limitation, I have related concerns about the reports of how many visits were completed by Health Visitors (herein referred to as “Specialist Public Health Nurses”) to Usual Care and FNP-visited families.	reveals a similar picture of excessive caseload (200-350 clients per HV, and up to 600 in some cases). For any one HV whose personal caseload is reduced by say one or two clients (ie across the local HV team) recruited to FNP, the opportunity to enhance support for their other clients is limited. So we agree with the reviewer that there is a risk, but that pragmatically it is likely to have a low impact. We feel that the current description provides balance on this topic and that readers will be able to make an informed view. Please see also our response to point 4 below (regarding some additional commentary added to Discussion).
2	In reporting characteristics of Usual Care in the current manuscript, it will be important to note that estimates of Midwife and Health-Visitor encounters were based upon participant reports (and subject to challenges with memory and uncertainty about how respondents interpreted questions regarding these encounters). Participants’ reports of encounters were based on interviews conducted at ages 6, 12, and 18 months postpartum but the FNP program goes through age 24 months. Note that other aspects of the interview conducted to measure Usual Care were conducted at 6, 12, 18, and 24 months postpartum.² Can the authors carry out estimates of Specialist Public Health Nurses (page 5, lines 57-59, and page 6, lines 4 and 5) by including data gathered at 24 months? (FNP encounters, as reported in the current paper, were based upon data from the FNP information system and go through 24 months.) These differences in sources of data need to be acknowledged in the current report given that FNP-visited and Usual-Care mothers reported nearly identical numbers of home visits. This seems odd given that FNP nurses were charged with assuming the Health Visitor role in the study sites.¹ What did the Specialized Public Health Nurses address in their relatively frequent home visits to families assigned to receive Health Visiting by FNP nurses? I tried to access the online supplement that laid out the questions examined in the “Usual-Care” paper² to address these questions, but the link is not working.	We already state that healthcare contacts were reported by trial participants. A common method was used for collecting participant reported healthcare encounters and this did not include an assessment at 24 months. Therefore, there is no opportunity to provide estimates of HV encounters based on data collected at 24 months. We have added further clarification that FNP visit were provided by family nurses. There are no data available on content of HV encounters. We have amended the reference to add in the DOI.
3	Given that FNP nurses assumed the Health-Visitor role, what did the Specialist nurses do with FNP-visited families in their homes? Weren’t they duplicating the work of the FNP nurses? In what way did they provide added service to the FNP-visited families? Did they spend the same amount of time when visiting Usual-Care and FNP-visited families?	We do not have data on this. It is possible that some self-reported HV contact for women in the intervention arm may actually relate to a FN visit instead (although this is rather unlikely given the relationship that would have existed between client and FN). It is possible too that HV contact may have occurred and been reported by women who had disengaged from the FNP programme (but similarly unlikely).

4	To some degree, the authors have addressed this inter-related set of concerns in the revised discussion section (page 23 lines 43-60 and page 24, lines 3-11). This revised discussion helps put this study in context, but it does not fully address the issues enumerated in 1 and 2 above.	We fully agree that it is valid to suggest that the open label nature of the trial may lead to some modification of service delivery for women in the control arm. Nevertheless, we consider that the potential for this to meaningfully occur is quite limited. We have highlighted this question in our discussion (and in previous work) and provided the data that are available to inform that discussion. In response to the reviewer's concern, we have added a further (a second) paragraph to expand on the question and to describe why we consider the actual risk here to be low. We have framed this within a contemporary behaviour change model (COM-B) as a means of exploring the potential for such bias to arise. We think this enables readers to understand the nature of the issue and assess the study and its result accordingly.
5	The current manuscript is an improvement over the previous version by giving some insight into the numbers and LOS of hospitalizations for injuries and ingestions by age (0-1 and >1-2 years) and by the thoughtful discussion of this issue in the discussion section. The data indicate that some admissions and discharges took place on the same day, however, suggesting that such admissions were not as serious as those that involved longer stays. A strong case can be made for excluding these less-serious cases (marked by same-day discharge.) The hypothesis, formulated on the basis of previous NFP trials, is that very young FNP-visited children were less likely to have been harmed due to problematic parental behavior. The analyses conducted so far appear to be by quantile regression, which truncates the range of values (and has included same-day discharges). A robust case can be made for analyzing days hospitalized in years 0-1 and years >1-2 (as well as combined as they were in the Memphis NFP trial) using Poisson regression given that extreme values have clinical meaning. In addition, the reader should be able to examine the specific ICD-10 codes associated with hospitalizations in the 0-2 age range (as the authors have done for all hospitalizations over the entire period in Table S8). Such analyses hold the potential for revealing in a much more objective way whether the program prevented serious harm to children at a highly vulnerable age -- when such harm is likely due to abuse or neglect and when it is likely to have life-long impacts. Given the enormous investments made in this study, the investigators have a responsibility to share these findings with the scientific and public- policy communities in determining the degree to which such effects are replicated.	We completed further exploratory analyses of these data. These have included all children admitted to hospital for an injury or ingestion rather than further excluding any (for example those apparently discharged on the same day). This is because the data available from NHS hospital records which shows an admission and discharge one day apart may not equate to one full day in hospital (ie the child may have been in hospital for only a few hours if the admission spanned midnight). This is a limitation of the available routine data. We have clarified that in the text. In addition, starting with a small number of admissions (49 in children aged under 1 year and 55 children aged 1 to under 2 years) and most being discharged in less than 24 hours, the resultant analysable dataset would be very small. The analysis previously presented was descriptive and did not use quantile regression. For children aged under 1 the median (25th to 75th quartile) length of stay (in days) were for the n=20 children in FNP group 0.75 (0.5 to 2) and for the n=29 children in the UC group 0.5 (0.5 to 2) with no difference observed (Poisson regression: Incidence rate ratio (IRR) 0.68, 95% CI 0.24 to 1.98, Ref = UC). For children aged between 1 and under 2 years old the median (25th to 75th centile) length of stay (in days) was for the n=21 admissions in FNP group 0.5 (0.5 to 1) and for the n=34 admissions in UC group 0.5 (0.5 to 1) with

		no observed difference (IRR 0.36, 95% CI 0.05 to 2.43, Ref = UC). This analysis provides evidence that the effects reported in the US trial are not replicated in the English trial cohort. We have added details of the above additional analysis and the admission codes for children aged under 2 to the existing Table S8 in the appendix.
6	The authors' response to my suggestion that they combine prenatal and 0-1 referrals to CIN in Table S6 was that there is no reason to combine pregnancy and 0-1 age categories because the data can be easily counted. Perhaps so, but the reader needs to have such data presented in a more accessible way. The 13 FNP cases in these combined age ranges (8.1%) compared to 5 UC (3.1%) is an important difference that will help with the interpretation of findings. Prenatal referrals probably reflect providers' concerns about profound mental illness, substance abuse, domestic violence, etc., conditions that are relatively unalterable and put the child in imminent harm. Such children need protection. Knowing the rate of referrals for CIN in these very earliest months is critical to revealing whether and how having FNP at this phase in life contributes to child protection/safety.	We have added the aggregate totals to the table as suggested (ie prenatal and 0-1 referrals).
7	In testing for surveillance bias in this trial, analyses could be conducted that would correspond to those conducted in the Elmira trial (reference 20), using the educational data from the BB2-6 study. Given differences in the health systems in Elmira (four decades ago) and the system in England today, use of health records for parallel analyses should be conducted with great circumspection. It's not clear that such data have the same meaning in the English contemporary context. Conducting these types of analyses using the educational data would be a service to those who fund this study and to the broader communities committed to improving maternal and child health.	The reviewer refers to subsequently reported analyses of the Elmira trial data to explore the potential of surveillance bias and which asked whether the lack of difference in that trial of rates of CPS determined maltreatment in the two years after the end of the NFP service delivery could be explained by actions of the nurses in identifying early potential problems (eg risk for maltreatment) and in linking those families up to other supportive health and human services. This would result in families with lower levels of concern being referred to CPS in the two-year period following the end of the programme than may be the case in the control arm. That analysis of all 56 children with state verified maltreatment in the first four years of life compared the intervention (NFP) and control groups on baseline maternal and 25-50 month child outcomes. We explored further the group of children referred to CPS and assessed as In Need (being the study primary outcome) by age 4 years old. As for the Elmira trial, we

had found no overall difference between trial arms for this outcome and using CIN as a dependent variable provided a moderate sized and numerically balanced sample for exploratory analysis (with n=60 and n=62 in FNP and UC groups respectively). It is also reasonable to hypothesise a similar mechanism of surveillance bias effect operating in relation to this outcome. We have interpreted the reviewer's reference to using educational data as being one of several potential indicators of (better / worse) child status rather than as the outcome of interest. This is because we did find differences between trial groups on early educational outcomes and there is no clear similar or direct mechanism whereby surveillance bias would affect this outcome.

Our analysis focused principally on baseline maternal differences to explore whether for children referred to CPS as assessed as In Need, there were systematic differences in family status such that children in one trial arm (eg FNP) were being referred at a lower level of overall concern. Mainly, we have not compared the two trial arms on subsequent outcomes (eg those occurring at any later time point) as some of these would have been assessed subsequent to a CPS referral and intervention, and therefore with potential to confound comparison (ie an effective intervention may have improved subsequent outcome rather than truly reflect a better / worse level of functioning at the point of referral). Nevertheless, we have also included in our comparison school readiness.

Therefore, we assume surveillance bias would operate in the way described in that hypothesised in the Elmira trial analysis and would be more evident in the referral of children for CIN (rather than CPP) and more likely to occur in the year or two after transition of care from FNP (ie from age 2 years, and not later than this) - which is also the age range evaluated in the Elmira exploratory analysis. Such surveillance / bias is not likely to impact upon educational outcomes measured at ages 4-7 years. These outcomes will have occurred after referral to CSC (and may be in large part driven by an intervention provided due to that referral) and therefore an unreliable marker of surveillance bias.

We have summarised these analyses in the new Table S18 in the appendix, along with some text in the Results section. There are mostly no baseline differences on maternal characteristics between trial arms for children referred and assessed as in need by age 4 years old. A greater proportion of children referred in the UC trial arm compared to the FNP trial arm had a mother who was not in education, employment or training at study entry.

		There are no differences between trial arms on the two measures of school readiness. We regard these analyses to provide some limited insight into the question of operation of surveillance bias in this study. Disentangling potential cause and effect (ie using outcomes assessed potentially after children were referred to indicate contemporary status) is particularly problematic for an outcome that can occur at any point during the study period. Interpreting any difference at school entry between trial arms would have been intriguing but inconclusive. The one difference found at baseline is not consistent with surveillance bias by the FNP nurse operating as predicted (and in fact runs counter) but similar to school readiness may also represents a distal marker for subsequent child status, although is at least better placed in the causal pathway.
8	Note on page 3, line 33, the impact of the Elmira program on maltreatment verified reports for the whole sample measured through child-age 15, current reference 30.	We have added a sentence to this effect and cited the reference.
9	Note also on page 3, line 33 the impact of NFP in Memphis on days hospitalized for injuries and ingestions through child-age 2, an objective possible indication of maltreatment. (Reference 2).	We have added a sentence to this effect and cited the reference.
10	My question about maternal educational achievement is this: Can the rates of missing data/records on maternal educational achievement be used to estimate the degree to which mothers completed educational milestones? Missing data may indicate that mothers were no longer in education when data were gathered, but the available data may reflect how much they achieved while they were enrolled. Those with no current records may hve dropped out of education. The authors have gathered these data even though some records are incomplete. Incomplete records may have educational meaning. Given the possible use of these data in estimating the effect of the program on this aspect of maternal functioning, it may be worth exploring along the lines I have suggested, or at least explaining whether this is possible.	This study gathered no new record of mother's educational outcomes (some maternally-reported data were collected and reported in the trial). The commissioning brief for the study allowed for child educational outcomes to be assessed only.
11	Note that length of visits was essentially identical to those found in US trials, that is 75-90 minutes. 3 References 1. https://eur03.safelinks.protection.outlook.com/?url=https%3A%2F%2Fbmcpediatr.biomedcentral.com%2Farticles%2F10.1186%2F1471-2431-13-114%23citeas&data=04%7C01%7Crobblingmr%40cardiff.ac.uk%7C64462c38baec47c053f808d9616b38d6%7Cbdb74b3095684856bdbf06759778fcbc%7C1%7C0%7C637647935420983759%7CUnknown%7CTWFpbGZsb3d8eyJWIjoiMC4wLjAwMDAiLCJQIjoiV2lu	We have added in a note describing the range of visit duration in the Elmira and Memphis trials and cited the reviewer's report accordingly.

MzliLCJBTil6lk1haWwiLCJXVCI6Mn0%3D%7C1000& sdata=GLqZNt7eJPcoEjcTXpsw7sC4%2Bzw8mA%2FKIaadEY0PbGk%3D& reserved=0 2. https://eur03.safelinks.protection.outlook.com/?url=https%3A%2F%2Fpubmed.ncbi.nlm.nih.gov%2F12387552%2F& data=04%7C01%7Crob lingmr%40cardiff.ac.uk%7C64462c38baec47c053f808d9616b38d6%7Cbdb74b3095684856dbf06759778fcbc%7C1%7C0%7C637647935420983759%7CUnknown%7CTWFpbGZsb3d8eyJWljo iMC4wLjAwMDAiLCJQljo iV2luMzliLCJBTil6lk1haWwiLCJXVCI6Mn0%3D%7C1000& sdata=aOiqSaHoX6T2Y9KZyCtmZ5V3OqdsS0qiuhi5ja492o%3D& reserved=0 3. https://eur03.safelinks.protection.outlook.com/?url=https%3A%2F%2Fpubmed.ncbi.nlm.nih.gov%2F12387552%2F& data=04%7C01%7Crob lingmr%40cardiff.ac.uk%7C64462c38baec47c053f808d9616b38d6%7Cbdb74b3095684856dbf06759778fcbc%7C1%7C0%7C637647935420983759%7CUnknown%7CTWFpbGZsb3d8eyJWljo iMC4wLjAwMDAiLCJQljo iV2luMzliLCJBTil6lk1haWwiLCJXVCI6Mn0%3D%7C1000& sdata=aOiqSaHoX6T2Y9KZyCtmZ5V3OqdsS0qiuhi5ja492o%3D& reserved=0	
--	--

VERSION 4 – REVIEW

REVIEWER	Olds, DL University of Colorado Denver, Department of Pediatrics I am the founder of Nurse-Family Partnership, the program tested in the Building Blocks trial and current follow-up. I am not an unbiased reviewer but have experience in leading three randomized clinical trials of the program in the US and reviewing other trials of the program in other contexts. My university receives royalties for the use of NFPO intellectual property, but I do not take the royalties as personal income above my university-determined salary. I periodically receive honoraria for speaking and writing and receive those honoraria as personal income.
REVIEW RETURNED	29-Nov-2021

GENERAL COMMENTS	I have proposed specific analyses that have strong scientific merit given results from previous trials of FNP. So far, the authors have not revealed the data or conducted analyses in sufficient detail to address these issues thoroughly. Here are my thoughts on this manuscript:  1. The authors list no limitations on page 2. I suggest that the issues listed below constitute limitations that they should acknowledge. 2. I remain concerned about the high number of visits paid to the Usual Care group. When FNP was embraced by the English government, those involved in Health Visiting leadership indicated that the maximum number of visits Health Visitors would be able to complete, given their high caseloads, would be three. This differs greatly from the mean number of visits reported in original report of this trial in The Lancet (mean=16.25 home visits), compared to an average of 8.60 visits reported for those assigned to the FNP condition. Since the original Lancet report, the authors published a paper (Usual Care) that indicates those assigned to Usual Care received an average of 5.01 home visits from Health Visitors plus 6.31 visits during clinic encounters. This compares to 4.70 home-visits from Health Visitors and 0.70 clinic-based visits for those assigned to FNP. The Usual Care report of SPHN visits has
---

been carried over to the current report that is under review here. The differences between the original Lancet report and numbers reported in the Usual Care paper are not explained. Please do so.

3. I remain concerned about the measurement of Health Visitors' contacts and their compensatory efforts to address the needs of children and families on their caseloads given that they knew which families had been assigned to the control group and that their service was being evaluated. This is a classic condition in which those serving Usual-Care participants are motivated, and in a position, to engage in "compensatory equalization." The revised discussion section addresses this and reduces this concern, but given the discrepancies in numbers of SPHN visits between the original Lancet report and the current one, the critical reviewer will want to see this issue reconsidered after the discrepancies between the two phases of follow-up that report on SPHN visits are reconciled. I address this issue along with other below with italicized text in response to the authors' response to my earlier review.

4. The Supplementary Appendix to the "Usual Care" article, along with its links to topics covered in the surveys, do not specify the questions posed to mothers to ascertain the frequency (or content) of Health Visitor visits. This is especially concerning given that the study design called for FNP nurses to take over the role of Health Visitors for those assigned to the FNP condition. Why did they visit so frequently given that they knew which families were visited by FNP nurses and which were in the Usual Care group? While the authors have addressed this issue in response to my raising this question in the last review, my concerns have not subsided, especially given the inconsistencies in reported number of visits completed by SPHN's. Clearly, this issue deserves greater explication of the data source and analysis as elaborated below in order to interpret the results of this trial and follow-up. Here are my specific suggestions:

a. Please explain the differences between the original report and the current one in quantification of Health Visitor encounters (now referred to as Specialist Public Health Nurses).

b. Please send a link to the questions posed to study participants to address this issue (along with the other questions related to SPHN encounters).

c. Please show the distributions of visits along with the means for each of the treatment conditions for each of the time periods in which these data were gathered – both for the original report (with a reported mean of 16.25 visits to the Usual Care group) and the more recent one.

d. The authors have examined in the Usual Care paper the degree to which SPHN's visited those at greater risk, but only after reducing the number of visits completed to dichotomous variables (< 4 versus 4+ visits). This way of reporting the data minimizes the differences between the high and lower risk groups. It will be helpful to the reader and scientific community to see the corresponding means and distributions re-analysed for the contrasting groups defined on the basis of risk, and for the numbers of visits reported in the original Lancet paper and the Usual Care paper. Relying on the 25th and 75th percentile ranges truncates differences.

e. Equally important, the analyses reported in the Usual Care paper do not show treatment differences in numbers of visits completed for those defined at risk (or not) by treatment condition. Testing for mean differences will help the reader interpret the absence of treatment differences found for those at greater risk. The analysis sums across the FNP and Usual Care conditions; it should be repeated separately for each treatment group, or even better could consist of a TX x Risk model with number of visits examined at the dependent variable.

5. There are major questions about the interpretation of referrals to child protection services. I'm not convinced, as the authors suggest, that enrollment in these services represents the degree to which children from each treatment condition were equally maltreated. There are several aspects to this, starting with the likelihood that FNP-visited families were referred to child protection earlier and at lower threshold of severity.

a. The authors have chosen not to report the number of referrals made to child protection that did not result in further action, but they have the data. They should include referrals in the current report. Referrals not leading to a CIN designation are likely to reflect efforts on the part of service providers (and others) concerned with

ensuring the child's wellbeing. While some of these referrals may come from sources without concern for the family or child's needs, most will. The reader should be shown the rates of referral by treatment condition and child age (prenatal, 0-< 1, etc.) as this will elaborate the degree to which and when families visited by FNP nurses versus Usual Care were engaged by the system to address children's and families' needs. FNP nurses work with health and social-care services to ensure that families' needs are addressed as soon as possible – just as Health Visitors do. One way they differ, however, is that FNP nurses visit more frequently and follow detailed protocols to address the range of maternal, child, and family health and developmental needs in pregnancy, infancy, and toddlerhood.

6. This also is likely to lead to higher rates of parental awareness of their young children's needs and subsequent engagement in the healthcare system to ensure that their children's injuries, ingestions, and other needs are addressed thoroughly through routine care, and, if needed, A&E. The manuscript needs to be written with a deeper appreciation for FNP nurses' encouraging families to make use of office-based care and A&E to address injuries and ingestions on the part of children. This is likely to lead to increases in office- and A&E-encounters for relatively minor conditions, but fewer days hospitalized for serious injuries and ingestions occurring in the first two years of life. In the original Lancet paper, the authors combined A&E attendance with hospital admissions for injuries and ingestions in their report of primary outcomes, which masks the way the program works. I will now turn to my earlier review and the authors' response to that review to elaborate this theme. Note that my response to their response is given in italics in the table below.

Response to reviewer comments (BB2 main manuscript to BMJOpen) Nov 2021
We thank the reviewers for all their time, expertise and comments on our manuscript and for facilitating its revision.

Review comment Author response

1 While it's true that the authors have consistently referred to this work as an "Open Label" trial, the usual meaning of Open Label is that participants and providers are aware of some specialized treatment like a drug or medical device that does not directly compete with the usual-care providers themselves in estimating their effectiveness. In the context of this study, the meaning of "Open" needs to be revealed more thoroughly, given that FNP nurses assumed the role of the Health Visitor in the study sites.¹ This reduced scope of Health Visitor practice in the FNP sites, put them in a position to visit those in "Usual Care" more frequently and with more focus than typically is possible for Health Visiting in the UK. While the authors have examined this issue in a paper they have published on this topic,² it is not clear whether usual care from Health Visitors in this study was representative of the way Health Visitors delivered the program to this population at the start of the trial. This feature of the design raises questions about its external validity, given that Usual Care was potentially augmented by having FNP nurses take over the role of Health Visitors for half of the targeted

population. While the current version of this manuscript goes further in acknowledging this limitation, I have related concerns about the reports of how many visits were completed by Health Visitors (herein referred to as "Specialist Public Health Nurses") to Usual Care and FNP- visited families. As the reviewer notes we provided added focus on this point in our revised manuscript including the potential for service professionals to enhance provision for woman allocated to the usual care arm (ie the implications of an open label approach).

The minimal reduction in client caseload for individual HVs in trial sites associated with some women entering FNP is unlikely to have afforded much opportunity to importantly enhance their offer. The comparative caseload figures we now provide talks to that but of course such data are imperfect in providing a definitive

comparison. However, our qualitative work at the time of the trial (eg focus group data from HVs at trial sites) reveals a similar picture of excessive caseload (200-350 clients per HV, and up to 600 in some cases). For any one HV whose personal caseload is reduced by say one or two clients (ie across the local HV team) recruited to FNP, the opportunity to enhance support for their other clients is limited.

Response: As noted above, I have new questions related to this issue that need to be addressed in resolving this issue. SPHN's have other clients with needs among the 200 to 600 they each serve. Are they able to increase the number of visits to others with needs as extensively as they appear to have for those in the Usual Care arm of this trial? I'm not convinced by the rationale that 1 or 2 cases with greater needs could be visited as frequently as reported either in the original Lancet paper or this one – especially when they were reported to have visited FNP assigned families so frequently.

So we agree with the reviewer that there is a risk, but that pragmatically it is likely to have a low impact. We feel that the current description provides balance on this topic and that readers will be able to make an informed view. Please see also our response to point 4 below (regarding some additional commentary added to Discussion).

2 In reporting characteristics of Usual Care in the current manuscript, it will be important to note that estimates of Midwife and Health-Visitor encounters were based upon participant reports (and subject to challenges with memory and uncertainty about how respondents interpreted questions regarding these encounters). Participants' reports of encounters were based on interviews conducted at ages 6, 12, and 18 months postpartum but the FNP program goes through age 24 months. Note that other aspects of the interview conducted to measure Usual Care were conducted at 6, 12, 18, and 24 months postpartum.² Can the authors carry out estimates of Specialist Public Health Nurses (page 5, lines 57-59, and page 6, lines 4 and 5) by including data gathered at 24 months? (FNP encounters, as reported in the current paper, were based upon data from the FNP information system and go through 24 months.) These differences in sources of data need to be acknowledged in the current report given that FNP- visited and Usual-Care mothers reported nearly identical numbers of home visits. This seems odd given that FNP nurses were charged with assuming the Health Visitor role in the study sites.¹ What did the Specialized Public Health Nurses address in their relatively frequent home visits to families assigned to receive Health Visiting by FNP nurses? I tried to access the online supplement that laid out the questions examined in the "Usual-Care" paper² to address these questions, but the link is not working.

As noted above, the supplemental appendix and the Usual Care paper do not provide linkages to the specific questions asked of participants to estimate the number of visits received by SPHN's and FNP nurses. The reader should be able to review these questions given the unusually large number reported in the original Lancet report and the smaller number reported in this paper. What exactly were the questions mothers answered? Why are the numbers reported in the two papers so different? We already state that healthcare contacts were reported by trial participants.

A common method was used for collecting participant reported healthcare encounters and this did not include an assessment at 24 months. Therefore, there is no opportunity to provide estimates of HV encounters based on data collected at 24 months.

We have added further clarification that FNP visit were provided by family nurses. There are no data available on content of HV encounters.

We have amended the reference to add in the DOI.

3 Given that FNP nurses assumed the Health-Visitor role, what did the Specialist nurses do with FNP-visited families in their homes? Weren't they duplicating the work of the FNP nurses? In what way did they provide added service to the FNP-visited families? Did they spend the same amount of time when visiting Usual-Care and FNP-visited families?

The Discussion section should reflect on the discrepancy between the large numbers of visits completed by SPHN's and the lack of clarity in their purpose when visiting FNP families, given that FNP's had been assigned the role of Health Visitors. Note again the inconsistency in reported number of visits competed in the original Lancet report and the current one. Why is this? We do not have data on this.

It is possible that some self-reported HV contact for women in the intervention arm may actually relate to a FN visit instead (although this is rather unlikely given the relationship that would have existed between client and FN). It is possible too that HV contact may have occurred and been reported by women who had disengaged from the FNP programme (but similarly unlikely).

4 To some degree, the authors have addressed this inter-related set of concerns in the revised discussion section (page 23 lines 43-60 and page 24, lines 3-11). This revised discussion helps put this study in context, but it does not fully address the issues enumerated in 1 and 2 above.

I appreciate the authors attempts to address this issue by adding the second paragraph explain why they think the risk is low for SPHNs to modify usual care, but I'm not convinced.

As noted above, given the enormous caseloads followed by heath visitors at the time of this trial, it is remarkable that they were able to visit control-group families either 16 times as reported in the Lancet paper, or to complete an average of 5 home visits and 6.3 clinic encounters as reported in the current paper. This means that on average, families in the control group were receiving 2-5 times as many visits as might be considered "usual care" even taking into account the special needs of adolescent mothers. This is insufficiently addressed in this study.

We fully agree that it is valid to suggest that the open label nature of the trial may lead to some modification of service delivery for women in the control arm. Nevertheless, we consider that the potential for this to meaningfully occur is quite limited. We have highlighted this question in our discussion (and in previous work) and provided the data that are available to inform that discussion.

In response to the reviewer's concern, we have added a further (a second) paragraph to expand on the question and to describe why we consider the actual risk here to be low. We have framed this within a contemporary behaviour change model (COM-B) as a means of exploring the potential for such bias to arise. We think this enables readers to understand the nature of the issue and assess the study and its result accordingly.

5 The current manuscript is an improvement over the previous version by giving some insight into the numbers and LOS of hospitalizations for injuries and ingestions by age (0-1 and >1-2 years) and by the thoughtful discussion of this issue in the discussion section. The data indicate that some admissions and discharges took place on the same day, however, suggesting that such admissions were not as serious as those that involved longer stays. A strong case can be made for excluding these less-serious cases (marked by same-day discharge.) The hypothesis, formulated on the basis of previous NFP trials, is that very young FNP-visited children were less likely to have been harmed due to problematic parental behavior. The analyses conducted so We completed further exploratory analyses of these data.

These have included all children admitted to hospital for an injury or ingestion rather than further excluding any (for example those apparently discharged on the same day). This is because the data available from NHS hospital records which shows an admission and discharge one day apart may not equate to one

far appear to be by quantile regression, which truncates the range of values (and has included same-day discharges). A robust case can be made for analysing days hospitalized in years 0-1 and years >1-2 (as well as combined as they were in the Memphis NFP trial) using Poisson regression given that extreme values have clinical meaning. In addition, the reader should be able to examine the specific ICD-10 codes associated with hospitalizations in the 0-2 age range (as the authors have done for all hospitalizations over the entire period in Table S8). Such analyses hold the potential for revealing in a much more objective way whether the program prevented serious harm to children at a highly vulnerable age -- when such harm is likely due to abuse or neglect and when it is likely to have life-long impacts. Given the enormous investments made in this study, the investigators have a responsibility to share these findings with the scientific and public-policy communities in determining the degree to which such effects are replicated.

Response: The analysis the authors have conducted does not address the question I have raised that calls for the exclusion of same-day discharges, given the relative minor conditions represented by these "admissions." These admissions likely reflect the kinds of conditions evaluated through A&E encounters.

In the interest of transparency, the authors should report the full actual distribution of hospitalization LOS for this young age group (<2) and analyse FNP-Usual Care differences in LOS in which discharges occurred on days following admissions. The authors give an example of a child being admitted on one date and discharged on another, but with short actual lengths of stay (measured in hours). This scenario will be relatively infrequently occurring compared to the scenario in which concerned parents take their children in for examination, may be admitted for a variety of reasons, but discharged shortly thereafter. It may be possible to identify those cases that the authors describe by examining every case that falls into the 1-day LOS category. Even if it is not possible to do so, removing the cases with the same date of admission and discharge will provide a clearer picture of admissions for serious injuries.

Serious, like-threatening injuries will lead to relatively long LOS's. And this outcome can be measured relatively objectively using the data the authors hold, and following the method I've recommended. The analysis the authors have conducted does not address the issue I've identified. They indicate that "the resultant dataset would be very small." Thank god for that. The solution to this is simply to show the distributions by treatment and conduct the analysis. The authors have a responsibility to conduct the analysis I'm suggesting, given its importance for science and policy. This is a simple request that will reveal whether FNP reduced an outcome of great clinical, public health, and scientific importance found in a previous trial. full day in hospital (ie the child may have been in hospital for only a few hours if the admission spanned midnight). This is a limitation of the available routine data. We have clarified that in the text. In addition, starting with a small number of admissions (49 in children aged under 1 year and 55 children aged 1 to under 2 years) and most being discharged in less than 24 hours, the resultant analysable dataset would be very small.

The analysis previously presented was descriptive and did not use quantile regression.

For children aged under 1 the median (25th to 75th quartile) length of stay (in days) were for the n=20 children in FNP group 0.75 (0.5 to 2) and for the n=29 children in the UC group 0.5 (0.5 to 2) with no difference observed (Poisson regression: Incidence rate ratio (IRR) 0.68, 95% CI 0.24 to 1.98, Ref = UC).

For children aged between 1 and under 2 years old the median (25th to 75th centile) length of stay (in days) was for the n=21 admissions in FNP group 0.5 (0.5 to 1) and for the n=34 admissions in UC group 0.5 (0.5 to 1) with no observed difference (IRR 0.36, 95% CI 0.05 to 2.43, Ref = UC).

This analysis provides evidence that the effects reported in the US trial are not replicated in the English trial cohort.

Consistent with my comments above, I need to repeat that what's missing in Table S8 is a careful delineation of the lengths of stay and the time periods over which the admissions took place. A&E encounters that lead to admissions are likely to take place when regular paediatric and primary care physician office hours are not open and parents are concerned about their children's wellbeing. Table S8 does not delineate the admissions sufficiently well to allow the reader to sort this out. While Table S9 addresses the timing of admissions, it does not sort out the numbers of days hospitalized by child-age or the diagnoses associated with those admissions. Pre-schoolers playing outside get hurt and there is no evidence from prior trials that NFP reduced hospitalizations for those reasons among older children. Admissions for head trauma and other serious injuries to a four-month-old are entirely different. Readers need to see the data displayed and analysed in ways that transparently reveal the pattern. The authors' response in Tables S8 and S9 provide more detail but simply do not address the issue I have raised. We have added details of the above additional analysis and the admission codes for children aged under 2 to the existing Table S8 in the appendix.

6 The authors' response to my suggestion that they combine prenatal and 0-1 referrals to CIN in Table S6 was that there is no reason to combine pregnancy and 0-1 age categories because the data can be easily counted. Perhaps so, but the reader needs to have such data presented in a more accessible way. The 13 FNP cases in these combined age ranges (8.1%) compared to 5 UC (3.1%) is an important difference that will help with the interpretation of findings. Prenatal referrals probably reflect providers' concerns about profound mental illness, substance abuse, domestic violence, etc., conditions that are relatively unalterable and put the child in imminent harm. Such children need protection. Knowing the rate of referrals for CIN in these very earliest months is critical to revealing whether and how having FNP at this phase in life contributes to child protection/safety.

Thank you. We have added the aggregate totals to the table as suggested (ie prenatal and 0-1 referrals).

7 In testing for surveillance bias in this trial, analyses could be conducted that would correspond to those conducted in the Elmira trial (reference 20), using the educational data from the BB2-6 study. Given differences in the health systems in Elmira (four decades ago) and the system in England today, use of health records for parallel analyses should be conducted with great circumspection. It's not clear that such data have the same meaning in the English contemporary context. Conducting these types of analyses using the educational data would be a service to those who fund this study and to the broader communities committed to improving maternal and child health.

The reviewer refers to subsequently reported analyses of the Elmira trial data to explore the potential of surveillance bias and which asked whether the lack of difference in that trial of rates of CPS determined maltreatment in the two years after the end of the NFP service delivery could be explained by actions of the nurses in identifying early potential problems (eg risk for maltreatment) and in linking those families up to other supportive health and human services. This would result in families with lower levels of concern being referred to CPS in the two-year period following the end of the programme than may be the case in the control arm.

That analysis of all 56 children with state verified maltreatment in the first four years of life compared the intervention (NFP) and control groups on baseline maternal and 25-50 month child outcomes.

We explored further the group of children referred to CPS and assessed as In Need (being the study primary outcome) by age 4 years old. As for the Elmira trial, we had found no overall difference between trial arms for this outcome and using CIN as a dependent variable provided a moderate sized and numerically balanced sample for exploratory analysis (with n=60 and n=62 in FNP and UC groups respectively). It is also reasonable to hypothesise a similar mechanism of surveillance bias effect operating in relation to this outcome. We have interpreted the reviewer's reference to using educational data as being one of several potential indicators of (better / worse) child status rather than as the outcome of interest. This is because we did find differences between trial groups on early educational outcomes and there is no clear similar or direct mechanism whereby surveillance bias would affect this outcome.

Our analysis focused principally on baseline maternal differences to explore whether for children referred to CPS as assessed as In Need, there were systematic differences in family status

such that children in one trial arm (eg FNP) were being referred at a lower level of overall concern. Mainly, we have not compared the two trial arms on subsequent outcomes (eg those occurring at any later time point) as some of these would have been assessed subsequent to a CPS referral and intervention, and therefore with potential to confound comparison (ie an effective intervention may have improved subsequent outcome rather than truly reflect a better / worse level of functioning at the point of referral).

Nevertheless, we have also included in our comparison school readiness.

Therefore, we assume surveillance bias would operate in the way described in that hypothesised in the Elmira trial analysis and would be more evident in the referral of children for CIN (rather than CPP) and more likely to occur in the year or two after transition of care from FNP (ie from age 2 years, and not later than this) - which is also the age range evaluated in the Elmira exploratory analysis. Such surveillance / bias is not likely to impact upon educational outcomes measured at ages 4-7 years. These outcomes will have occurred after referral to CSC (and may be in large part driven by an intervention provided due to that referral) and therefore an unreliable marker of surveillance bias.

Response: Thank you. The inclusion of baseline characteristics helps with the analysis. FNP-visited mothers with a CIN were at lower baseline risk than their Usual Care counterparts; that is, they were less likely to be NEET at registration, a pattern consistent with FNP nurses' greater surveillance of mothers' and families' needs. This is consistent with their children being in "Looked-After" care for two fewer months than their Usual-Care counterparts. Moreover, I would suggest the following: examine LOS hospitalized for injuries and ingestions in the first two years of life – after removing cases admitted and discharged on the same day. While the numbers will be small, a case can be made for exploring whether Usual Care and NFP Children in Need have the same number of days hospitalized for injuries and ingestions in the first 2 years of life. In addition, I was surprised to see that the analysis did not include Key Stage 1 assessments. To round out the picture, the analysis should include these outcomes – after adjusting for birth month. These outcomes are objective measures of child health and development and will help with the interpretation of the child protective service/CIN outcomes.

We have summarised these analyses in the new Table S18 in the appendix, along with some text in the Results section. There are mostly no baseline differences on

maternal characteristics between trial arms for children referred and assessed as in need by age 4 years old. A greater proportion of children referred in the UC trial arm compared to the FNP trial arm had a mother who was not in education, employment or training at study entry. There are no differences between trial arms on the two measures of school readiness.

We regard these analyses to provide some limited insight into the question of operation of surveillance bias in this study. Disentangling potential cause and effect (ie using outcomes assessed potentially after children were referred to indicate contemporary status) is particularly problematic for an outcome that can occur at any point during the study period. Interpreting any difference at school entry between trial arms would have been intriguing but inconclusive. The one difference found at baseline is not consistent with surveillance bias by the FNP nurse operating as predicted (and in fact runs counter) but similar to school readiness may also represent a distal marker for subsequent child status, although is at least better placed in the causal pathway.

8 Note on page 3, line 33, the impact of the Elmira program on maltreatment verified reports for the whole sample measured through child-age 15, current reference 30.

Thanks We have added a sentence to this effect and cited the reference.

9 Note also on page 3, line 33 the impact of NFP in Memphis on days hospitalized for injuries and ingestions through child-age 2, an objective possible indication of maltreatment. (Reference 2).

Thanks We have added a sentence to this effect and cited the reference.

10 My question about maternal educational achievement is this: Can the rates of missing data/records on maternal educational achievement be used to estimate the degree to which mothers completed educational milestones? Missing data may indicate that mothers were no longer in education when data were gathered, but the available data may reflect how much they achieved while they were enrolled. Those with no current records may have dropped out of education. The authors have gathered these data even though some records are incomplete. Incomplete records may have educational meaning. Given the possible use of these data in estimating the effect of the program on this aspect of maternal functioning, it may be worth exploring along the lines I have suggested, or at least explaining whether this is possible.

OK This study gathered no new record of mother's educational outcomes (some maternally- reported data were collected and reported in the trial). The commissioning brief for the study allowed for child educational outcomes to be assessed only.

11 Note that length of visits was essentially identical to those found in US trials, that is 75-90 minutes. 3 References

1.

[https://eur03.safelinks.protection.outlook.com/?url=https%3A%2F%2Fbmcpediatr.biomedcentral.com%2Farticles%2F10.1186%2F1471-2431-13-114%23citeas&data=04%7C01%7Crobmingmr%40cardiff.ac.uk%7C64462c38baec47c053f808d9616b38d6%7Cbdb74b3095684856bdbf06759778fcbc%7C1%7C0%7C637647935420983759](https://eur03.safelinks.protection.outlook.com/?url=https%3A%2F%2Fbmcpediatr.biomedcentral.com%2Farticles%2F10.1186%2F1471-2431-13-114%23citeas&data=04%7C01%7Crobmingmr%40cardiff.ac.uk%7C64462c38baec47c053f808d9616b38d6%7Cbdb74b3095684856bdbf06759778fcbc%7C1%7C0%7C637647935420983759%7CUnknown%7CTWFPbGZsb3d8eyJWljoiMC4wLjAwMDAiLCJQIjoiV2luMzliLCJBTil6lk1haWwiL)

[%7CUnknown%7CTWFPbGZsb3d8eyJWljoiMC4wLjAwMDAiLCJQIjoiV2luMzliLCJBTil6lk1haWwiL](https://eur03.safelinks.protection.outlook.com/?url=https%3A%2F%2Fbmcpediatr.biomedcentral.com%2Farticles%2F10.1186%2F1471-2431-13-114%23citeas&data=04%7C01%7Crobmingmr%40cardiff.ac.uk%7C64462c38baec47c053f808d9616b38d6%7Cbdb74b3095684856bdbf06759778fcbc%7C1%7C0%7C637647935420983759%7CUnknown%7CTWFPbGZsb3d8eyJWljoiMC4wLjAwMDAiLCJQIjoiV2luMzliLCJBTil6lk1haWwiL)

[CJXVCI6Mn0%3D%7C1000&sdata=GLqZNt7eJPcoEjcTXpsw7sC4%2Bzw8mA%2FKlaadEY0PbGk%3D&reserved=0](https://eur03.safelinks.protection.outlook.com/?url=https%3A%2F%2Fbmcpediatr.biomedcentral.com%2Farticles%2F10.1186%2F1471-2431-13-114%23citeas&data=04%7C01%7Crobmingmr%40cardiff.ac.uk%7C64462c38baec47c053f808d9616b38d6%7Cbdb74b3095684856bdbf06759778fcbc%7C1%7C0%7C637647935420983759%7CUnknown%7CTWFPbGZsb3d8eyJWljoiMC4wLjAwMDAiLCJQIjoiV2luMzliLCJBTil6lk1haWwiL)

2.

<https://eur03.safelinks.protection.outlook.com/?url=https%3A%2F%2Fbmjopen.bmj.com>

	%2Fcontent%2F8%2F5%2Fe020152&data=04%7C01%7Croblingmr%40cardiff.ac.uk%7C64 462c38baec47c053f808d9616b38d6%7Cbdb74b3095684856bdbf06759778fcbc%7C1%7C0%7C 637647935420983759%7CUnknown%7CTWFpbGZsb3d8eyJWljojMC4wLjAwMDAiLCJQIjoiV2lu MzliLCJBTiI6Ik1haWwiLCJXVCI6Mn0%3D%7C1000&sdata=Pdb9%2FJwP8t6LYZXo SEXswwE aLcHiy07kJeV7EkBI3TM%3D&reserved=0 3. https://eur03.safelinks.protection.outlook.com/?url=https%3A%2F%2Fpubmed.ncbi.nlm.nih.gov%2F12387552%2F&data=04%7C01%7Croblingmr%40cardiff.ac.uk%7C64462c38baec47c053f808d9616b38d6%7Cbdb74b3095684856bdbf06759778fcbc%7C1%7C0%7C637647935420983759%7CUnknown%7CTWFpbGZsb3d8eyJWljojMC4wLjAwMDAiLCJQIjoiV2luMzliLCJBTiI6Ik1haWwiLCJXVCI6Mn0%3D%7C1000&sdata=aOiqaSaHoX6T2Y9KZyCtmZ5V3OqdsS0qiuhi5 ja492o%3D&reserved=0 3. We have added in a note describing the range of visit duration in the Elmira and Memphis trials and cited the reviewer's report accordingly. nih.gov%2F12387552%2F&data=04%7C01%7Croblingmr%40cardiff.ac.uk%7C64462c38baec47c053f808d9616b38d6%7Cbdb74b3095684856bdbf06759778fcbc%7C1%7C0%7C637647935420983759%7CUnknown%7CTWFpbGZsb3d8eyJWljojMC4wLjAwMDAiLCJQIjoiV2luMzliLCJBTiI6Ik1haWwiLCJXVCI6Mn0%3D%7C1000&sdata=aOiqaSaHoX6T2Y9KZyCtmZ5V3OqdsS0qiuhi5 ja492o%3D&reserved=0 Thanks
--	---

VERSION 4 – AUTHOR RESPONSE

#	Review comment	Author response
1	The authors list no limitations on page 2. I suggest that the issues listed below constitute limitations that they should acknowledge.	We have added bullet points regarding the data related to service contacts and data limitations inherent with using routine service data. We agree that these should be acknowledged.
2	I remain concerned about the high number of visits paid to the Usual Care group. When FNP was embraced by the English government, those involved in Health Visiting leadership indicated that the maximum number of visits Health Visitors would be able to complete, given their high caseloads, would be three. This differs greatly from the mean number of visits reported in original report of this trial in The Lancet (mean=16.25 home visits), compared to an average of 8.60 visits reported for those assigned to the FNP condition. Since the original Lancet report, the authors published a paper (Usual Care) that indicates those assigned to Usual Care received an average of 5.01 home visits from Health Visitors plus 6.31 visits during clinic encounters. This compares to 4.70 home-visits from Health Visitors and 0.70 clinic-based visits for those assigned to FNP. The	We are grateful to the reviewer for highlighting this inconsistency and for the opportunity to clarify the data. The original trial report contains two tables summarising health visitor contacts. In chapter 11 (which reports on Usual care) Table 11.5 (page 278) shows the number of contacts reported by women in the usual care trial arm, including at the three postpartum follow-up points where these questions were asked (at six, twelve and eighteen months postpartum) and disaggregated by those at home or in the clinic. The figures presented here are consistent with those presented in the BMJOpen publication on usual care referred to by the reviewer.

	Usual Care report of SPHN visits has been carried over to the current report that is under review here. The differences between the original Lancet report and numbers reported in the Usual Care paper are not explained. Please do so.	The second table (Table 14.7) from the report chapter on economic analysis provides a summary based on imputed data to account for the non-collection of a range of resource data at 24 months (which was also reported in the cited Lancet paper). This imputation strategy was applied generically across a range of services. The service delivery of public health functions identifies five points at which health visitors (or FNP nurses) may be scheduled to support parents, including one 'By 1 year' and the next one 'By 2-2.5 years'. Therefore, an assumption that visits would have been maintained at a similar frequency in the last six months of the trial period as prior to likely to be highly conservative (overestimate). The service specification for the Healthy Child Programme (HCP) identified five scheduled points for delivery specifically by health visitors rather than the three mentioned by the reviewer. Actions: We have noted in the discussion section the differences between reported rates of health visitor contacts.
3	I remain concerned about the measurement of Health Visitors' contacts and their compensatory efforts to address the needs of children and families on their caseloads given that they knew which families had been assigned to the control group and that their service was being evaluated. This is a classic condition in which those serving Usual-Care participants are motivated, and in a position, to engage in "compensatory equalization." The revised discussion section addresses this and reduces this concern, but given the discrepancies in numbers of SPHN visits between the original Lancet report and the current one, the critical reviewer will want to see this issue reconsidered after the discrepancies between the two phases of follow-up that report on SPHN visits are reconciled. I address this issue along with other below with italicized text in response to the authors' response to my earlier review.	We have clarified the inconsistency referred to by the reviewer above.
4	The Supplementary Appendix to the "Usual Care" article, along with its links to topics covered in the surveys, do not specify the questions posed to mothers to ascertain the frequency (or content) of Heath Visitor visits. This is especially concerning given that the	4a – this is addressed above

study design called for FNP nurses to take over the role of Health Visitors for those assigned to the FNP condition. Why did they visit so frequently given that they knew which families were visited by FNP nurses and which were in the Usual Care group? While the authors have addressed this issue in response to my raising this question in the last review, my concerns have not subsided, especially given the inconsistencies in reported number of visits completed by SPHN's. Clearly, this issue deserves greater explication of the data source and analysis as elaborated below in order to interpret the results of this trial and follow-up. Here are my specific suggestions: a. Please explain the differences between the original report and the current one in quantification of Health Visitor encounters (now referred to as Specialist Public Health Nurses). b. Please send a link to the questions posed to study participants to address this issue (along with the other questions related to SPHN encounters). c. Please show the distributions of visits along with the means for each of the treatment conditions for each of the time periods in which these data were gathered – both for the original report (with a reported mean of 16.25 visits to the Usual Care group) and the more recent one. d. The authors have examined in the Usual Care paper the degree to which SPHN's visited those at greater risk, but only after reducing the number of visits completed to dichotomous variables (< 4 versus 4+ visits). This way of reporting the data minimizes the differences between the high and lower risk groups. It will be helpful to the reader and scientific community to see the corresponding means and distributions re-analysed for the contrasting groups defined on the basis of risk, and for the numbers of visits reported in the original Lancet paper and the Usual Care paper. Relying on the 25th and 75th percentile ranges truncates differences. e. Equally important, the analyses reported in the Usual Care paper do not show treatment differences in numbers of visits completed for those defined at risk (or not) by treatment condition. Testing for mean differences will help the reader interpret the absence of treatment differences found for those at greater risk. The analysis sums across the FNP and Usual Care conditions; it should be repeated separately for each treatment group, or even better could consist of a TX x Risk model	4b – we have included the relevant sections of the postpartum CRFs in section XII of the revised appendices. 4c – following clarification of the data described above, we refer the reader to the summary data presented in table 2 of the cited 'What is usual care ...' paper which reports these figures. 4d – such a re-analysis of data previously published in 'usual care' is beyond the scope of the current paper. Further analysis of such data may be of interest we agree but also may also be informed by a variety of approaches and not solely that which we used or suggested by the reviewer. 4e – see response for 4d
--	---

	with number of visits examined at the dependent variable.	
5	There are major questions about the interpretation of referrals to child protection services. I'm not convinced, as the authors suggest, that enrollment in these services represents the degree to which children from each treatment condition were equally maltreated. There are several aspects to this, starting with the likelihood that FNP-visited families were referred to child protection earlier and at lower threshold of severity. The authors have chosen not to report the number of referrals made to child protection that did not result in further action, but they have the data. They should include referrals in the current report. Referrals not leading to a CIN designation are likely to reflect efforts on the part of service providers (and others) concerned with ensuring the child's wellbeing. While some of these referrals may come from sources without concern for the family or child's needs, most will. The reader should be shown the rates of referral by treatment condition and child age (prenatal, 0-< 1, etc.) as this will elaborate the degree to which and when families visited by FNP nurses versus Usual Care were engaged by the system to address children's and families' needs. FNP nurses work with health and social-care services to ensure that families' needs are addressed as soon as possible – just as Health Visitors do. One way they differ, however, is that FNP nurses visit more frequently and follow detailed protocols to address the range of maternal, child, and family health and developmental needs in pregnancy, infancy, and toddlerhood.	We do report on the number of children ever referred to CSCS (n=206, n=208; FNP:UC). Of these, we then report on those who are additionally assigned a Child Protection Plan (n=52, n=49; FNP:UC) and Looked after – a subset of those with CPP (n=25, n=27) as well as Child In Need (the primary outcome). Given that for each outcome (CIN, CPP, CLA) the numbers are virtually equivalent between trial arms, as are the number ever referred it is unclear what additional useful information about referral resulting in no further action (ie CIN, CPP, CLA) is being sought. We have focused on referrals for children assessed as CIN for which we do present data on age at referral by trial arm. The numbers of children referred and assessed as CIN before age 2 years is small and equivalent across trial arms (NB numbers suppressed for children aged under 1 years). Further presentation by age of children referred to CSCS but not assessed as requiring additional support (eg CIN, CPP, CLA) will involve very small numbers, and which would not reportable (due to the small numbers policy restrictions) and would not reveal anything about maltreatment outcomes differences between trial arms as it specifically focuses on events / outcomes considered not considered actionable by CSCS.
6	This also is likely to lead to higher rates of parental awareness of their young children's needs and subsequent engagement in the healthcare system to ensure that their children's injuries, ingestions, and other needs are addressed thoroughly through routine care, and, if needed, A&E. The manuscript needs to be written with a deeper appreciation for FNP nurses' encouraging families to make use of office-based care and A&E to address injuries and ingestions on the part of children. This is likely to lead to increases in office- and A&E-encounters for relatively minor conditions, but fewer days	We already address this in the Discussion section (eg paragraph four of the section on Strengths and weaknesses of the study in which we describe the theoretical and operational nature of FNP nurse practice. We do agree that this an important point to make, however. In interviews with health visitors / SPHNs at the time of the trial and feedback subsequently, would note that staff in existing usual care services shared this ethos (with their FNP colleagues) and saw FNP work as high-quality health visiting practice.

	hospitalized for serious injuries and ingestions occurring in the first two years of life. In the original Lancet paper, the authors combined A&E attendance with hospital admissions for injuries and ingestions in their report of primary outcomes, which masks the way the program works.	
	I will now turn to my earlier review and the authors' response to that review to elaborate this theme. Note that my response to their response is given in italics in the table below.	NB In the copy of the reviewer's comments supplied, no structure / formatting was applied, so we have applied this in the central column to distinguish previous and new comments.
7	Previous comment: While it's true that the authors have consistently referred to this work as an "Open Label" trial, the usual meaning of Open Label is that participants and providers are aware of some specialized treatment like a drug or medical device that does not directly compete with the usual-care providers themselves in estimating their effectiveness. In the context of this study, the meaning of "Open" needs to be revealed more thoroughly, given that FNP nurses assumed the role of the Health Visitor in the study sites.¹ This reduced scope of Health Visitor practice in the FNP sites, put them in a position to visit those in "Usual Care" more frequently and with more focus than typically is possible for Health Visiting in the UK. While the authors have examined this issue in a paper they have published on this topic,² it is not clear whether usual care from Health Visitors in this study was representative of the way Health Visitors delivered the program to this population at the start of the trial. This feature of the design raises questions about its external validity, given that Usual Care was potentially augmented by having FNP nurses take over the role of Health Visitors for half of the targeted population. While the current version of this manuscript goes further in acknowledging this limitation, I have related concerns about the reports of how many visits were completed by Health Visitors (herein referred to as "Specialist Public Health Nurses") to Usual Care and FNP-visited families. As the reviewer notes we provided added focus on this point in our revised manuscript including the potential for service professionals to enhance provision for woman allocated to the usual care arm (ie the implications of an open label approach). Author response (partial): The minimal reduction in client caseload for individual HVs in trial sites associated with some	We recognise that a limitation in our trial data was that lack of detail captured for each health visitor contact and that we were reliant upon maternal reported data (as data were not routinely available via health service records on such provision as would have been the case for FNP visits). In responding to the reviewer's comments above we have pointed to the data included in our 'Usual care ...' paper which describes the number of contacts women in the control arm reported receiving by 18 months. These amounted to an approximate total of 5.5 home visits (and 6.4 clinic contacts). We noted above that the service specification for HCP identifies a suggested 5 scheduled opportunities for specific HV contact (but which is not a fidelity goal). The degree to which the reported level of HV contact represents an extensive increase on normally provided care is open to question, we would suggest. Our previous paper on usual care found some support for the care provision being enhanced for families with greater need and this does fit the relevant service model of progressive support to families with greater levels of need. Home-, rather than clinic-, based visits are also likely to both offer more substantive opportunity to support to mothers but will also be considerably more time / resource intensive. Clinic-based contacts are more likely to be relatively fleeting and have few opportunities for the depth and focus provided via FNP delivered home visits. If there was indeed an extensively higher level of support provided by health visitors to women in

	women entering FNP is unlikely to have afforded much opportunity to importantly enhance their offer. The comparative caseload figures we now provide talks to that but of course such data are imperfect in providing a definitive comparison. However, our qualitative work at the time of the trial (eg focus group data from HVs at trial sites) reveals a similar picture of excessive caseload (200-350 clients per HV, and up to 600 in some cases). For any one HV whose personal caseload is reduced by say one or two clients (ie across the local HV team) recruited to FNP, the opportunity to enhance support for their other clients is limited. Response: As noted above, I have new questions related to this issue that need to be addressed in resolving this issue. SPHN's have other clients with needs among the 200 to 600 they each serve. Are they able to increase the number of visits to others with needs as extensively as they appear to have for those in the Usual Care arm of this trial? I'm not convinced by the rationale that 1 or 2 cases with greater needs could be visited as frequently as reported either in the original Lancet paper or this one – especially when they were reported to have visited FNP assigned families so frequently.	the usual care arm because HVs were motivated to deliver enhanced care for women identified to them as in the non-intervention trial arm then this still may only represent best usual practice for a service operating (as it should) a progressive universal approach. We have mapped out in the Discussion section a model describing the opportunities and requirements for this to have occurred and presented the data available to us to help inform that consideration. This question is addressed now in considerable detail in this amended manuscript and readers can themselves consider the value of those perspectives.
8	Previous comment: In reporting characteristics of Usual Care in the current manuscript, it will be important to note that estimates of Midwife and Health-Visitor encounters were based upon participant reports (and subject to challenges with memory and uncertainty about how respondents interpreted questions regarding these encounters). Participants' reports of encounters were based on interviews conducted at ages 6, 12, and 18 months postpartum but the FNP program goes through age 24 months. Note that other aspects of the interview conducted to measure Usual Care were conducted at 6, 12, 18, and 24 months postpartum.² Can the authors carry out estimates of Specialist Public Health Nurses (page 5, lines 57-59, and page 6, lines 4 and 5) by including data gathered at 24 months? (FNP encounters, as reported in the current paper, were based upon data from the FNP information system and go through 24 months.) These differences in sources of data need to be	We have included details of the specific items used in the postpartum telephone surveys in the revised appendices document. The discrepancy in reported figures was addressed above.

	acknowledged in the current report given that FNP- visited and Usual-Care mothers reported nearly identical numbers of home visits. This seems odd given that FNP nurses were charged with assuming the Health Visitor role in the study sites.¹ What did the Specialized Public Health Nurses address in their relatively frequent home visits to families assigned to receive Health Visiting by FNP nurses? I tried to access the online supplement that laid out the questions examined in the "Usual-Care" paper² to address these questions, but the link is not working. Author response: We already state that healthcare contacts were reported by trial participants. A common method was used for collecting participant reported healthcare encounters and this did not include an assessment at 24 months. Therefore, there is no opportunity to provide estimates of HV encounters based on data collected at 24 months. We have added further clarification that FNP visit were provided by family nurses. There are no data available on content of HV encounters. Response: As noted above, the supplemental appendix and the Usual Care paper do not provide linkages to the specific questions asked of participants to estimate the number of visits received by SPHN's and FNP nurses. The reader should be able to review these questions given the unusually large number reported in the original Lancet report and the smaller number reported in this paper. What exactly were the questions mothers answered? Why are the numbers reported in the two papers so different?	
9	Previous comment: Given that FNP nurses assumed the Health-Visitor role, what did the Specialist nurses do with FNP-visited families in their homes? Weren't they duplicating the work of the FNP nurses? In what way did they provide added service to the FNP-visited families? Did they spend the same amount of time when visiting Usual-Care and FNP-visited families? Authors response: We do not have data on this. It is possible that some self-reported HV	Our response regarding the discrepancy in figures is addressed above. The limitations of the data available to us has been addressed in the revised Discussion section.

	contact for women in the intervention arm may actually relate to a FN visit instead (although this is rather unlikely given the relationship that would have existed between client and FN). It is possible too that HV contact may have occurred and been reported by women who had disengaged from the FNP programme (but similarly unlikely). Response: The Discussion section should reflect on the discrepancy between the large numbers of visits completed by SPHN's and the lack of clarity in their purpose when visiting FNP families, given that FNP's had been assigned the role of Health Visitors. Note again the inconsistency in reported number of visits completed in the original Lancet report and the current one. Why is this?	
10	Previous comment: To some degree, the authors have addressed this inter-related set of concerns in the revised discussion section (page 23 lines 43-60 and page 24, lines 3-11). This revised discussion helps put this study in context, but it does not fully address the issues enumerated in 1 and 2 above. Author's response: We fully agree that it is valid to suggest that the open label nature of the trial may lead to some modification of service delivery for women in the control arm. Nevertheless, we consider that the potential for this to meaningfully occur is quite limited. We have highlighted this question in our discussion (and in previous work) and provided the data that are available to inform that discussion. In response to the reviewer's concern, we have added a further (a second) paragraph to expand on the question and to describe why we consider the actual risk here to be low. We have framed this within a contemporary behaviour change model (COM-B) as a means of exploring the potential for such bias to arise. We think this enables readers to understand the nature of the issue and assess the study and its result accordingly. Response: I appreciate the authors attempts to address this issue by adding the second paragraph explain why they think the risk is low for SPHNs to modify usual care, but I'm not convinced. As noted above, given the	In responding to the reviewer's valid comments on this topic we have provided a framework to consider the feasibility and likelihood of there being substantive enhancement of usual care. This is to better enable all readers to form a judgement on this point, even if that is not convincing for this reviewer. Furthermore, we have clarified the extant service framework for health visitor mediated delivery of the Healthy Child Programme at the time of the trial and which indicates a greater number of visits than the reviewer had been advised about. It should also be noted that all trial sites (comprising partnerships between local authorities and health boards) were only eligible for the trial if they demonstrated organisational commitment to support new families (as determined by Dept of Health in the piloting and commissioning of FNP in England) and the progressive universal model for HCP delivery which intends to enhance support to those most at need. Providing such enhanced support to all families is not expected, only for those with additional needs. So good quality usual care would not mean that all families on the health visitor list require an equal number of enhanced / extra visits.

	enormous caseloads followed by health visitors at the time of this trial, it is remarkable that they were able to visit control-group families either 16 times as reported in the Lancet paper, or to complete an average of 5 home visits and 6.3 clinic encounters as reported in the current paper. This means that on average, families in the control group were receiving 2-5 times as many visits as might be considered “usual care” even taking into account the special needs of adolescent mothers. This is insufficiently addressed in this study.	
11	Previous comment: The current manuscript is an improvement over the previous version by giving some insight into the numbers and LOS of hospitalizations for injuries and ingestions by age (0-1 and >1-2 years) and by the thoughtful discussion of this issue in the discussion section. The data indicate that some admissions and discharges took place on the same day, however, suggesting that such admissions were not as serious as those that involved longer stays. A strong case can be made for excluding these less-serious cases (marked by same-day discharge.) The hypothesis, formulated on the basis of previous NFP trials, is that very young FNP-visited children were less likely to have been harmed due to problematic parental behavior. The analyses conducted so far appear to be by quantile regression, which truncates the range of values (and has included same-day discharges). A robust case can be made for analyzing days hospitalized in years 0-1 and years >1-2 (as well as combined as they were in the Memphis NFP trial) using Poisson regression given that extreme values have clinical meaning. In addition, the reader should be able to examine the specific ICD-10 codes associated with hospitalizations in the 0-2 age range (as the authors have done for all hospitalizations over the entire period in Table S8). Such analyses hold the potential for revealing in a much more objective way whether the program prevented serious harm to children at a highly vulnerable age -- when such harm is likely due to abuse or neglect and when it is likely to have life-long impacts. Given the enormous investments made in this study, the investigators have a responsibility to share these findings with the scientific and public-policy communities in determining the degree to which such effects are replicated.	As we have previously reported in response to the reviewer’s recommendations, with the routine hospital admission data available to us it is not possible to confidently assume that an admission and discharge one day apart is in fact any longer than a reported same day discharge (for reasons that were clarified in the previously re-submitted manuscript). This is a limitation that we have previously acknowledged. If it was advisable to remove from analysis children who were discharged within a short space of time (ie as a reasonable proxy for a presentation of less serious concern) that should in theory include some children currently listed as being discharged one day apart. While the reviewer suggests that a case-by-case analysis may allow for a determination of a 1-day LOS, that is not in fact the case, in part because it would involve extrapolating with some uncertainty from clinical diagnostic codes. However, we have amended the data summarised in table 9 to additionally show the number of children by age group (under 1 year, 1 to under 2 years, 25 months to under 6 years) admitted to hospital for an injury or an ingestion and discharged either nominally on the same day (ie admission = discharge date) or after 1 day or more (for children older than 25 months, we present those discharged on same day, after 1 day and after 2 days or more). These selected categorisations were chosen to allow values to be shown in outputs (ie permitted within the small numbers policy). Finer categorisation would have been restricted and the small numbers involved would not have been informative. The rates of longer stays (ie one or more days is the same across trial arms (FNP: n=10, 50%; UC: n=14, 48.3%) for children under 1 years old. This pattern is repeated for children aged 1 to under 2 years (FNP: n=7, 33.3%; UC: n=11, 32.4%) and aged 25 months to under 6 years (for

Author response: We completed further exploratory analyses of these data. These have included all children admitted to hospital for an injury or ingestion rather than further excluding any (for example those apparently discharged on the same day). This is because the data available from NHS hospital records which shows an admission and discharge one day apart may not equate to one full day in hospital (ie the child may have been in hospital for only a few hours if the admission spanned midnight). This is a limitation of the available routine data. We have clarified that in the text. In addition, starting with a small number of admissions (49 in children aged under 1 year and 55 children aged 1 to under 2 years) and most being discharged in less than 24 hours, the resultant analysable dataset would be very small. The analysis previously presented was descriptive and did not use quantile regression. For children aged under 1 the median (25th to 75th quartile) length of stay (in days) were for the n=20 children in FNP group 0.75 (0.5 to 2) and for the n=29 children in the UC group 0.5 (0.5 to 2) with no difference observed (Poisson regression: Incidence rate ratio (IRR) 0.68, 95% CI 0.24 to 1.98, Ref = UC). For children aged between 1 and under 2 years old the median (25th to 75th centile) length of stay (in days) was for the n=21 admissions in FNP group 0.5 (0.5 to 1) and for the n=34 admissions in UC group 0.5 (0.5 to 1) with no observed difference (IRR 0.36, 95% CI 0.05 to 2.43, Ref = UC). This analysis provides evidence that the effects reported in the US trial are not replicated in the English trial cohort. We have added details of the above additional analysis and the admission codes for children aged under 2 to the existing Table S8 in the appendix. Response: The analysis the authors have conducted does not address the question I have raised that calls for the exclusion of same-day discharges, given the relative minor conditions represented by these “admissions.” These admissions likely reflect the kinds of conditions evaluated through A&E encounters.	Discharge after 1 day, FNP: n=18, 27.6%; UC: n=17, 30.4%; for Discharge after 2 days or more, FNP: n=11, 16.9%; UC: n=10, 17.9%). We have added these data to Table S9.
--	--

In the interest of transparency, the authors should report the full actual distribution of hospitalization LOS for this young age group (<2) and analyse FNP-Usual Care differences in LOS in which discharges occurred on days following admissions. The authors give an example of a child being admitted on one date and discharged on another, but with short actual lengths of stay (measured in hours). This scenario will be relatively infrequently occurring compared to the scenario in which concerned parents take their children in for examination, may be admitted for a variety of reasons, but discharged shortly thereafter. It may be possible to identify those cases that the authors describe by examining every case that falls into the 1-day LOS category. Even if it is not possible to do so, removing the cases with the same date of admission and discharge will provide a clearer picture of admissions for serious injuries.

Serious, like-threatening injuries will lead to relatively long LOS's. And this outcome can be measured relatively objectively using the data the authors hold, and following the method I've recommended. The analysis the authors have conducted does not address the issue I've identified. They indicate that "the resultant dataset would be very small." Thank god for that. The solution to this is simply to show the distributions by treatment and conduct the analysis. The authors have a responsibility to conduct the analysis I'm suggesting, given its importance for science and policy. This is a simple request that will reveal whether FNP reduced an outcome of great clinical, public health, and scientific importance found in a previous trial.

Consistent with my comments above, I need to repeat that what's missing in Table S8 is a careful delineation of the lengths of stay and the time periods over which the admissions took place. A&E encounters that lead to admissions are likely to take place when regular paediatric and primary care physician office hours are not open and parents are concerned about their children's wellbeing. Table S8 does not delineate the admissions sufficiently well to allow the reader to sort this out. While Table S9 addresses the timing of admissions, it does not sort out the numbers of days hospitalized by child-age or the diagnoses associated with those admissions. Pre-schoolers playing outside get hurt and there is no evidence from prior trials that NFP reduced hospitalizations for those

	reasons among older children. Admissions for head trauma and other serious injuries to a four-month-old are entirely different. Readers need to see the data displayed and analysed in ways that transparently reveal the pattern. The authors' response in Tables S8 and S9 provide more detail but simply do not address the issue I have raised.	
12	Previous comment: The authors' response to my suggestion that they combine prenatal and 0-1 referrals to CIN in Table S6 was that there is no reason to combine pregnancy and 0-1 age categories because the data can be easily counted. Perhaps so, but the reader needs to have such data presented in a more accessible way. The 13 FNP cases in these combined age ranges (8.1%) compared to 5 UC (3.1%) is an important difference that will help with the interpretation of findings. Prenatal referrals probably reflect providers' concerns about profound mental illness, substance abuse, domestic violence, etc., conditions that are relatively unalterable and put the child in imminent harm. Such children need protection. Knowing the rate of referrals for CIN in these very earliest months is critical to revealing whether and how having FNP at this phase in life contributes to child protection/safety. Author response: We have added the aggregate totals to the table as suggested (ie prenatal and 0-1 referrals). Reviewer response: Thank you	No response requested
13	Previous comment: In testing for surveillance bias in this trial, analyses could be conducted that would correspond to those conducted in the Elmira trial (reference 20), using the educational data from the BB2-6 study. Given differences in the health systems in Elmira (four decades ago) and the system in England today, use of health records for parallel analyses should be conducted with great circumspection. It's not clear that such data have the same meaning in the English contemporary context. Conducting these types of analyses using the educational data would be a service to those who fund this study and to the broader communities committed to improving maternal and child health.	We have modified the description of these results in the manuscript and also draw the parallel to the findings from the reviewer's own work accordingly. This introduces and supports the notion that FN nurses are (intentionally) practising differently and that this has may have consequential effects for the interpretation of some outcomes. We have not included KS1 data here because such an outcome is potentially impacted upon by an intervention put in place to address the child's CIN status (rather than being, for example, an indication of baseline or contemporary risk). Where we have previously included early years

Author response: The reviewer refers to subsequently reported analyses of the Elmira trial data to explore the potential of surveillance bias and which asked whether the lack of difference in that trial of rates of CPS determined maltreatment in the two years after the end of the NFP service delivery could be explained by actions of the nurses in identifying early potential problems (eg risk for maltreatment) and in linking those families up to other supportive health and human services. This would result in families with lower levels of concern being referred to CPS in the two-year period following the end of the programme than may be the case in the control arm. That analysis of all 56 children with state verified maltreatment in the first four years of life compared the intervention (NFP) and control groups on baseline maternal and 25-50 month child outcomes. We explored further the group of children referred to CPS and assessed as In Need (being the study primary outcome) by age 4 years old. As for the Elmira trial, we had found no overall difference between trial arms for this outcome and using CIN as a dependent variable provided a moderate sized and numerically balanced sample for exploratory analysis (with n=60 and n=62 in FNP and UC groups respectively). It is also reasonable to hypothesise a similar mechanism of surveillance bias effect operating in relation to this outcome. We have interpreted the reviewer's reference to using educational data as being one of several potential indicators of (better / worse) child status rather than as the outcome of interest. This is because we did find differences between trial groups on early educational outcomes and there is no clear similar or direct mechanism whereby surveillance bias would affect this outcome. Our analysis focused principally on baseline maternal differences to explore whether for children referred to CPS as assessed as In Need, there were systematic differences in family status such that children in one trial arm (eg FNP) were being referred at a lower level of overall concern. Mainly, we have not compared the two trial arms on subsequent outcomes (eg those occurring at any later time point) as some of these would have been assessed subsequent to a CPS referral and intervention, and therefore with potential	outcome data we have highlighted the caution to be exercised in interpretation. As mentioned above, the absolute distinction between same day or next day discharge as marker of less clinically important admissions is unreliable. Even if we included these data, the total number of children in the study sample aged under 2 years, admitted to hospital for an injury or ingestion and discharged after one or more days is small (as suggested / suspected by the reviewer) - eg for the FNP trial arm this is n=17 (of all 760 children) and the analysis itself is focused only on the 60 children assessed as in need (eg in this trial arm). In addition, the number of admissions will reduce even further when admission occurring after the referral and assessment as CIN are omitted (as LOS following admission for injury/ingestion may be temporally confounded with referral and assessment as CIN). Among children requiring social services involvement, reduced availability of social workers at night, weekend and over bank holidays can substantially extend admission durations, whilst engagement with FNP may reduce duration of stay due to staff confidence the child can be closely monitored in their home environment. It is not possible to quantify these effects, but they have been confirmed as plausible by senior clinical based paediatricians. A general consideration we have is the very large number of comparative analyses now undertaken in the originally agreed analysis plan (which itself indicated exploratory analyses) and subsequently suggested via the review process. While we initially nominated a primary outcome to address this, there has been no further adjustment made to compensate for the number of statistical tests being undertaken and for which we could be criticised for. Formally applying a conservative correction (eg such as Bonferroni for all secondary outcomes) would drastically reduce / change the picture; and is the reason we have avoided any suggestion of statistical trends and are wary about further exploratory analyses which increasingly sub-categorise the sample or restrict data. Some exploratory analyses of course is reasonable and necessary but any such analysis can be subject to over-interpretation by readers regardless of stated caveats. Nevertheless (as noted above), the possibility of surveillance bias (consistent with the cited analysis from the US trials) is affirmed by the present analysis.
---	---

to confound comparison (ie an effective intervention may have improved subsequent outcome rather than truly reflect a better / worse level of functioning at the point of referral). Nevertheless, we have also included in our comparison school readiness.

Therefore, we assume surveillance bias would operate in the way described in that hypothesised in the Elmira trial analysis and would be more evident in the referral of children for CIN (rather than CPP) and more likely to occur in the year or two after transition of care from FNP (ie from age 2 years, and not later than this) - which is also the age range evaluated in the Elmira exploratory analysis. Such surveillance / bias is not likely to impact upon educational outcomes measured at ages 4-7 years. These outcomes will have occurred after referral to CSC (and may be in large part driven by an intervention provided due to that referral) and therefore an unreliable marker of surveillance bias.

We have summarised these analyses in the new Table S18 in the appendix, along with some text in the Results section. There are mostly no baseline differences on maternal characteristics between trial arms for children referred and assessed as in need by age 4 years old. A greater proportion of children referred in the UC trial arm compared to the FNP trial arm had a mother who was not in education, employment or training at study entry. There are no differences between trial arms on the two measures of school readiness.

We regard these analyses to provide some limited insight into the question of operation of surveillance bias in this study. Disentangling potential cause and effect (ie using outcomes assessed potentially after children were referred to indicate contemporary status) is particularly problematic for an outcome that can occur at any point during the study period. Interpreting any difference at school entry between trial arms would have been intriguing but inconclusive. The one difference found at baseline is not consistent with surveillance bias by the FNP nurse operating as predicted (and in fact runs counter) but similar to school readiness may also represent a distal marker for

	subsequent child status, although is at least better placed in the causal pathway. Reviewer response: Thank you. The inclusion of baseline characteristics helps with the analysis. FNP-visited mothers with a CIN were at lower baseline risk than their Usual Care counterparts; that is, they were less likely to be NEET at registration, a pattern consistent with FNP nurses' greater surveillance of mothers' and families' needs. This is consistent with their children being in "Looked-After" care for two fewer months than their Usual-Care counterparts. Moreover, I would suggest the following: examine LOS hospitalized for injuries and ingestions in the first two years of life – after removing cases admitted and discharged on the same day. While the numbers will be small, a case can be made for exploring whether Usual Care and NFP Children in Need have the same number of days hospitalized for injuries and ingestions in the first 2 years of life. In addition, I was surprised to see that the analysis did not include Key Stage 1 assessments. To round out the picture, the analysis should include these outcomes – after adjusting for birth month. These outcomes are objective measures of child health and development and will help with the interpretation of the child protective service/CIN outcomes.	
14	Reviewer comment: Note on page 3, line 33, the impact of the Elmira program on maltreatment verified reports for the whole sample measured through child-age 15, current reference 30. Author response: We have added a sentence to this effect and cited the reference. Reviewer response: Thank you.	No response required
15	Reviewer comment: Note also on page 3, line 33 the impact of NFP in Memphis on days hospitalized for injuries and ingestions through child-age 2, an objective possible indication of maltreatment. (Reference 2).	No response required

	Author response: We have added a sentence to this effect and cited the reference. Reviewer response: Thank you	
16	Reviewer comment: My question about maternal educational achievement is this: Can the rates of missing data/records on maternal educational achievement be used to estimate the degree to which mothers completed educational milestones? Missing data may indicate that mothers were no longer in education when data were gathered, but the available data may reflect how much they achieved while they were enrolled. Those with no current records may have dropped out of education. The authors have gathered these data even though some records are incomplete. Incomplete records may have educational meaning. Given the possible use of these data in estimating the effect of the program on this aspect of maternal functioning, it may be worth exploring along the lines I have suggested, or at least explaining whether this is possible. Author response: This study gathered no new record of mother's educational outcomes (some maternally- reported data were collected and reported in the trial). The commissioning brief for the study allowed for child educational outcomes to be assessed only. Reviewer response: OK	No response indicated
17	Reviewer comment: Note that length of visits was essentially identical to those found in US trials, that is 75-90 minutes. 3 References 1. https://eur03.safelinks.protection.outlook.com/?url=https%3A%2F%2Fbmcpediatr.biomedcentral.com%2Farticles%2F10.1186%2F1471-2431-13-114%23citeas&data=04%7C01%7Crobblingm%40cardiff.ac.uk%7C64462c38baec47c053f808d9616b38d6%7Cbdb74b3095684856bdf06759778fcbc%7C1%7C0%7C637647935420983759 %7CUnknown%7CTWFpbGZsb3d8eyJWIjoiMC4wLjAwMDAiLCJQIjoiV2luMzIiLCJBTiI6Ik	No response indicated

1haWwiL CJXVCI6Mn0%3D%7C1000&sdata=GLqZNt 7eJPcoEjcTXpsw7sC4%2Bzw8mA%2FKlaa dEY0Pb Gk%3D&reserved=0 2. https://eur03.safelinks.protection.outlook.com /?url=https%3A%2F%2Fbmjopen.bmj.com %2Fcontent%2F8%2F5%2Fe020152&data= 04%7C01%7Crobtingmr%40cardiff.ac.uk%7 C64 462c38baec47c053f808d9616b38d6%7Cbd b74b3095684856bdbf06759778fcbc%7C1% 7C0%7C 637647935420983759%7CUnknown%7CTW FpbGZsb3d8eyJWljoimC4wLjAwMDAiLCJQI joiV2lu MzliLCJBTil6lk1haWwiLCJXVCI6Mn0%3D% 7C1000&sdata=Pdb9%2FJwP8t6LYZXoSEX swwE aLcHiy07kJeV7EkBI3TM%3D&reserved=0 3. https://eur03.safelinks.protection.outlook.com /?url=https%3A%2F%2Fpubmed.ncbi.nlm Author response: We have added in a note describing the range of visit duration in the Elmira and Memphis trials and cited the reviewer's report accordingly. Reviewer response: -	
---	--